# Comprehensive fitness landscape of SARS-CoV-2 Mᵖʳᵒ reveals insights into viral resistance mechanisms

**Julia M Flynn[1]***, **Neha Samant[1]**, **Gily Schneider-Nachum[1]**, **David T Barkan[2]**, **Nese Kurt Yilmaz[1]**, **Celia A Schiffer[1]**, **Stephanie A Moquin[2]**, **Dustin Dovala[2]**, **Daniel NA Bolon[1]***

[1]Department of Biochemistry and Molecular Biotechnology, University of Massachusetts Chan Medical School, Worcester, United States; [2]Novartis Institutes for Biomedical Research, Emeryville, United States

**Abstract** With the continual evolution of new strains of severe acute respiratory syndrome coronavirus-2 (SARS-CoV-2) that are more virulent, transmissible, and able to evade current vaccines, there is an urgent need for effective anti-viral drugs. The SARS-CoV-2 main protease (Mᵖʳᵒ) is a leading target for drug design due to its conserved and indispensable role in the viral life cycle. Drugs targeting Mᵖʳᵒ appear promising but will elicit selection pressure for resistance. To understand resistance potential in Mᵖʳᵒ, we performed a comprehensive mutational scan of the protease that analyzed the function of all possible single amino acid changes. We developed three separate high throughput assays of Mᵖʳᵒ function in yeast, based on either the ability of Mᵖʳᵒ variants to cleave at a defined cut-site or on the toxicity of their expression to yeast. We used deep sequencing to quantify the functional effects of each variant in each screen. The protein fitness landscapes from all three screens were strongly correlated, indicating that they captured the biophysical properties critical to Mᵖʳᵒ function. The fitness landscapes revealed a non-active site location on the surface that is extremely sensitive to mutation, making it a favorable location to target with inhibitors. In addition, we found a network of critical amino acids that physically bridge the two active sites of the Mᵖʳᵒ dimer. The clinical variants of Mᵖʳᵒ were predominantly functional in our screens, indicating that Mᵖʳᵒ is under strong selection pressure in the human population. Our results provide predictions of mutations that will be readily accessible to Mᵖʳᵒ evolution and that are likely to contribute to drug resistance. This complete mutational guide of Mᵖʳᵒ can be used in the design of inhibitors with reduced potential of evolving viral resistance.

*For correspondence:
Julia.Flynn@umassmed.edu
(JMF);
Dan.Bolon@umassmed.edu
(DNAB)

## Editor's evaluation

This manuscript utilizes modern molecular tools to construct a fitness landscape in SARS-CoV-2, yielding insight into potential resistance mechanisms. The paper is rigorous, well-written, and has very clear implications in the biomedical realm.

## Introduction

The COVID-19 pandemic, caused by the severe acute respiratory syndrome coronavirus-2 (SARS-CoV-2), has had an unprecedented impact on global health, the world economy, and our overall way of life. Despite the rapid deployment of mRNA and traditional vaccines against SARS-CoV-2, which have served to greatly improve patient outcomes and decrease spread of the disease, vaccines remain unavailable in many parts of the world and there is hesitancy to get vaccinated among portions of

the population. Additionally, the virus appears to be evolving mutations in the spike protein that reduce immune protection from both vaccines and prior infections. Additional strategies including direct-acting antiviral drugs are needed to combat the SARS-CoV-2 pandemic. The main protease (M^pro) of SARS-CoV-2 is a promising target for drug development with many laboratories working collaboratively to develop drugs against this protease, leading to thousands of M^pro inhibitors in the pipeline and the first FDA-authorized clinical drug against this target, Paxlovid. The use of drugs that target M^pro will apply selection pressure for the evolution of resistance. There is potential to design drugs with reduced likelihood of developing M^pro resistance, but these efforts will require an in-depth understanding of the evolutionary potential of the protease.

SARS-CoV-2 is a highly contagious virus responsible for the ongoing COVID-19 pandemic. SARS-CoV-2 belongs to the group of coronaviruses and has a positive-sense single-stranded RNA genome (*Macnaughton and Madge, 1978*). Immediately upon entry into the host cell, the SARS-CoV-2 virus translates its replicase gene (ORF1) into two overlapping large polyproteins produced in tandem by a ribosomal frameshift, pp1a and pp1ab (*Herold et al., 1993*). These polyproteins are cleaved by two cysteine proteases, M^pro (also known as the chymotrypsin-like protease, 3CL^pro or Nsp5) and the papain-like protease (PL^pro) to yield functional replication machinery indispensable to viral replication (*Ziebuhr et al., 1995*; *Lim et al., 2000*). The M^pro initiates autoproteolysis from the pp1a and pp1ab polypeptides at its N- and C-terminus, through a poorly understood mechanism (*Hsu et al., 2005b*). Subsequently, mature M^pro cuts at 11 additional cleavage sites in both pp1a and pp1ab (*Fan et al., 2004*). All the sites cut by M^pro include a conserved Gln at the P1 position, a small amino acid (Ser, Ala, or Gly) at the P1's position, and a hydrophobic residue (Leu, Phe, or Val) at the P2 position (*Hegyi et al., 2002*; *Thiel et al., 2003*). Along with its vital role in the liberation of viral proteins, M^pro also cleaves specific host proteins, an activity which has been shown to enhance viral replication (*Meyer et al., 2021*). Through its substrates, M^pro function is required for almost every known step in the viral life cycle.

M^pro is a highly attractive target for drug development against SARS-CoV-2 and future coronavirus-mediated pandemics for numerous reasons. M^pro plays an essential functional role in the viral life cycle so that blocking its function will impair viral propagation. M^pro is highly conserved among all coronaviruses making it likely that inhibitors will have broad efficacy in potential future pandemics. There are no human M^pro homologs, and it shares no overlapping substrate specificity with any known human protease, minimizing the possibility of side effects. Additionally, its nucleophilic cysteine active site enables the design of covalent inhibitors that provide advantages such as increased potency, selectivity, and duration of inhibition (*Singh et al., 2011*). For these reasons, M^pro has become one of the most characterized SARS-CoV-2 drug targets (*Jin et al., 2020*; *Zhang et al., 2020*; *Biering et al., 2021*; *Fischer et al., 2021*).

Native M^pro is a homodimer, and each monomer is composed of three domains (*Jin et al., 2020*). Domain I (8–101) and Domain II (102–184) are composed of antiparallel β-barrel structures. Cys145 and His41 make up M^pro's non-canonical catalytic dyads and are located in clefts between Domains I and II. Domain III (201–303) is an all α-helical domain that coordinates M^pro dimerization, which is essential for M^pro function (*Tan et al., 2005*). Much of the structural and enzymatic knowledge of SARS-CoV-2 M^pro has been derived from studies of SARS-CoV-1 that caused the 2003 SARS outbreak (*Ksiazek et al., 2003*), as well as MERS-CoV that caused the 2012 MERS outbreak (*Zaki et al., 2012*). M^pro from SARS-CoV-1 and SARS-CoV-2 differ in sequence at only 12 residues; however, SARS-CoV-2 M^pro exhibits increased structural flexibility and plasticity (*Bzówka et al., 2020*; *Estrada, 2020*; *Kneller et al., 2020*).

We performed comprehensive mutational analysis of SARS-CoV-2 M^pro to provide functional and structural information to aid in the design of effective inhibitors against the protease. Systematic mutational scanning assesses the consequences of all point mutations in a gene providing a comprehensive picture of the relationship between protein sequence and function (*Hietpas et al., 2011*; *Fowler and Fields, 2014*). Mutational scanning requires a selection step that separates variants based on function. Following selection, the frequency of each variant is assessed by deep sequencing to estimate functional effects. The resulting protein fitness landscape describes how all individual amino acid changes in a protein impact function and provides a detailed guide to the biophysical and biochemical properties that underlie fitness. Protein fitness landscapes identify mutation-tolerant positions that may readily contribute to drug resistance. These studies also elucidate mutation-sensitive residues

that are critical to function, making them attractive target sites for inhibitors with reduced likelihood of developing resistance. The work described here focuses on fitness landscapes without drug pressure because these provide critical information regarding Mpro mechanism and evolutionary potential that we hope will be useful in the efforts to combat SARS-CoV-2. We are pursuing investigations in the presence of inhibitors, but these experiments will require further optimization steps to make our yeast-based assays compatible with inhibition. Of note, mutational scans of other drug targets including lactamases (*Deng et al., 2012*; *Firnberg et al., 2014*) and oncogenes (*Choi et al., 2014*; *Ma et al., 2017*) have demonstrated the potential to accurately identify and predict clinically relevant resistance evolution.

In this study, we used systematic mutational scanning to analyze the functional effects of every individual amino acid change in Mpro. We developed three orthogonal screens in yeast to separate Mpro variants based on function. The first screen measures Mpro activity via loss of fluorescence resonance energy transfer (FRET) from a genetically-encoded FRET pair linked by the Nsp4/5 cleavage sequence (*Figure 1a*). The second screen similarly measures cleavage of the Nsp4/5 cut site; however, in this screen Mpro cleavage leads to inactivation of a transcription factor (TF) driving GFP expression (*Figure 1b*). The final screen leverages the toxicity of wild-type (WT) Mpro to yeast that is likely due to cleavage of essential yeast proteins, and leads to depletion of active variants during growth (*Figure 1c*). Following selection in the three screens, populations were subjected to deep sequencing in order to quantify function based on the enrichment or depletion of each variant.

We found that the functional scores between screens were correlated, indicating that they all captured key biophysical properties governing function. Our functional scores also correlated well with previously measured catalytic rates of purified individual mutants. Additionally, substitutions in Mpro from coronaviruses distantly related to SARS-CoV-2 consistently exhibited high function in our screens indicating that similar biophysical properties underlie the function of genetically diverse Mpro sequences. Our study revealed mutation-sensitive sites distal to the active site and dimerization interface. These sites reveal important communication networks that may be targeted by inhibitors. Our results provide a comprehensive dataset which can be used to design molecules with decreased vulnerability to resistance, by building drug-protein interactions at mutation-sensitive sites while avoiding mutation-tolerant residues.

## Results

### Expression of mature WT Mpro in yeast

The Mpro of SARS-CoV-2 is produced by self-cleavage of polyproteins translated from the viral RNA genome, and its enzymatic activity is inhibited by the presence of additional N- and C-terminal amino acids (*Xue et al., 2007*). To express Mpro with its authentic N-terminal serine residue, we generated a ubiquitin (Ub)-Mpro fusion protein. In yeast and other eukaryotes, Ub fusion proteins are cleaved by Ub-specific proteases directly C-terminal to the Ub, revealing the N-terminal residue of the fused protein, regardless of sequence (*Bachmair et al., 1986*). Expression of functionally active Mpro is toxic to yeast cells (*Alalam et al., 2021*). To control the expression level of Mpro while limiting its toxic side effects, we placed Ub-Mpro under control of the inducible and engineered LexA-ER-AD TF (*Ottoz et al., 2014*). LexA-ER-AD is a fusion of the bacterial LexA DNA binding protein, the human estrogen receptor (ER) and the B112 activation domain (AD), and its activity is tightly and precisely regulated by the hormone β-estradiol. We inserted 4 *lexA* boxes recognized by the LexA DNA binding domain (DBD) upstream of Ub-Mpro to control its expression. The Western blot in *Figure 1—figure supplement 1a* illustrates both induction of Mpro by β-estradiol and successful removal of the Ub moiety, indicating that the protease is being expressed in its mature and functional form. We performed a titration curve with β-estradiol to determine the lowest concentration at which Mpro can be expressed without inhibiting yeast cell growth while still enabling measurement of substrate cleavage (*Figure 1—figure supplement 1b*).

### Engineering of functional screens to monitor intracellular Mpro activity

We developed three distinct yeast screens to characterize the effects of Mpro variants on function (*Figure 1*). The first screen utilized a FRET-based reporter of two fluorescent proteins, YPet and CyPet, fused together with the Nsp4/5 MproCS cleavage site engineered in the middle (YPet-MproCS-CyPet)

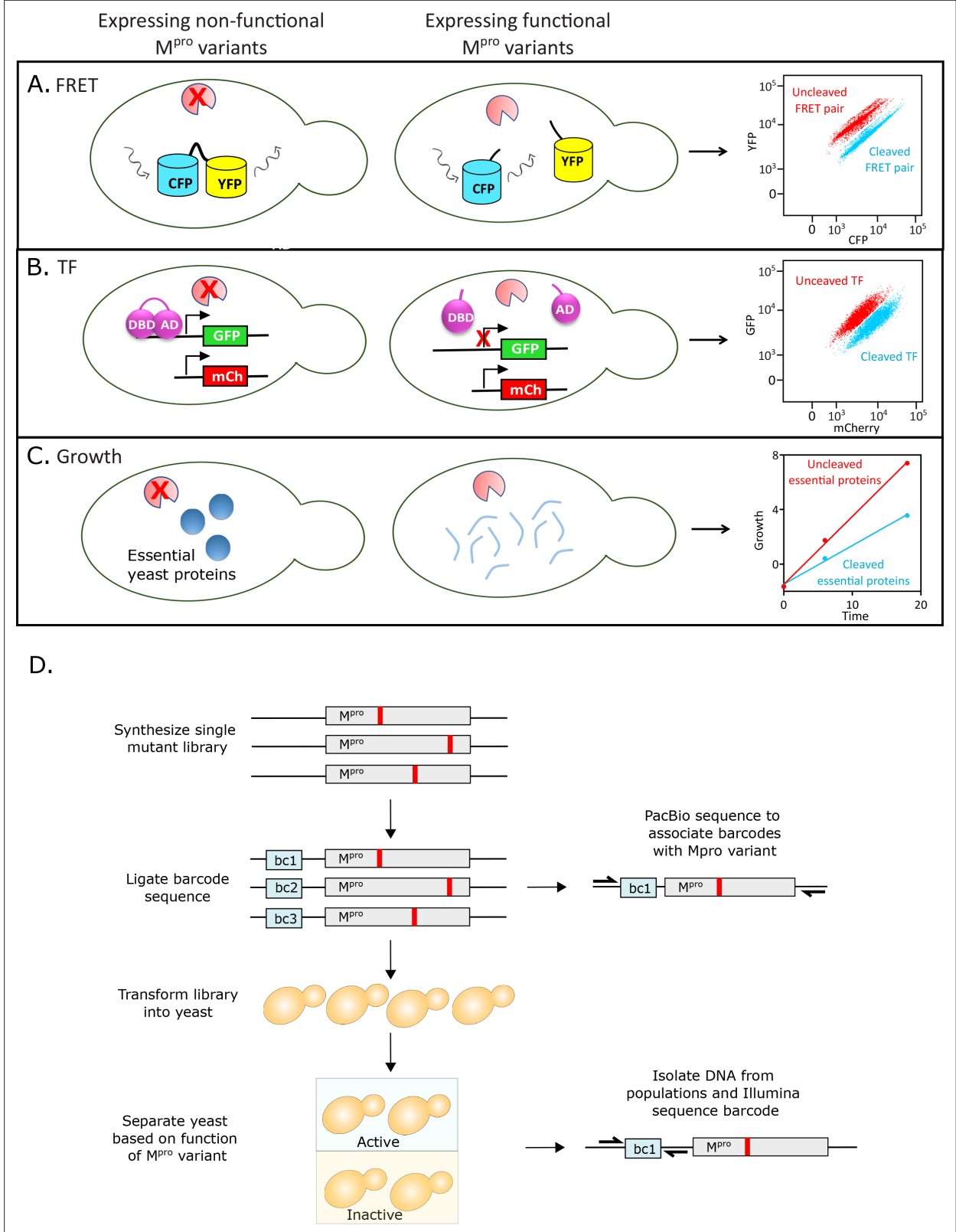

**Figure 1.** Experimental strategy to measure the function of all individual mutations of main protease (Mpro). (**A**) Fluorescence resonance energy transfer (FRET)-based reporter screen. The Mpro variants were sorted based on their ability to cleave at the Mpro cut-site, separating the YFP-CFP FRET pair. Cells were separated by fluorescence-activated single cell sorting (FACS) into cleaved (low FRET) and uncleaved (high FRET) populations. (**B**) Split transcription factor screen. Mpro variants were sorted based on their ability to cleave at the Mpro cut-site, separating the DNA binding domain (DBD) and

*Figure 1 continued on next page*

*Figure 1 continued*

activation domain (AD) of the Gal4 transcription factor. The transcription factor drives GFP expression from a galactose promoter. Cells were separated by FACS into cleaved (low GFP expression) and uncleaved (high GFP expression) populations. (**C**) Growth screen. Yeast cells expressing functional M^pro variants that cleave essential yeast proteins grow slowly and are depleted in bulk culture, while yeast cells expressing non-functional M^pro variants are enriched. (**D**) Barcoding strategy to measure frequency of all individual mutations of M^pro in a single experiment.

The online version of this article includes the following figure supplement(s) for figure 1:

**Figure supplement 1.** Main protease (M^pro) expression in cells harboring the LexA-UbM^pro plasmid construct.

(*Figure 1a*). The YPet-CyPet pair are derivatives of the YFP-CFP proteins that have been fluorescently optimized by directed evolution for intracellular FRET (*Nguyen and Daugherty, 2005*) and provide a 20-fold signal change upon cleavage. The linker between the two fluorescent proteins contains the M^pro cleavage site, TSAVLQ|SGFRK, the cut-site at the N-terminus of the M^pro. This is the most commonly used cut-site for in vitro cleavage assays, which allowed us to directly compare our mutational results to those that were previously published. One advantage of this assay is that the fluorescent readout directly reports on cleavage of a specific cut-site. The plasmid containing Ub-M^pro under the control of β-estradiol was transformed into yeast cells expressing a chromosomally integrated copy of YPet-M^proCS-CyPet. Expression of WT M^pro led to a β-estradiol-dependent decrease in FRET signal as measured by fluorescence-activated single cell sorting (FACS). Mutation of the essential catalytic cysteine of M^pro to alanine (C145A) abolished this change in FRET signal indicating that the change in signal was dependent on the presence of functional M^pro (*Figure 1—figure supplement 1c*).

The second screen utilized the DBD and AD of the Gal4 TF, separated by the Nsp4/5 cut site (*Johnston et al., 1986*; *Murray et al., 1993*). We used this engineered TF to drive GFP expression, enabling cells with varying levels of M^pro protease activity to be separated by FACS (*Figure 1b*). One benefit of this system is its signal amplification, as one cut TF can cause a reduction of more than one GFP molecule. However, due to this amplification, the fluorescent signal is indirectly related to cutting efficiency. Expression of Ub-M^pro in cells engineered with the split TF exhibited a β-estradiol-dependent decrease in GFP reporter activity that required the presence of catalytically functional M^pro protein (*Figure 1—figure supplement 1d*). The final screen leverages the toxicity of M^pro expression in yeast, which likely results from cleavage of essential yeast proteins by the protease (*Alalam et al., 2021*; *Figure 1c*). Increasing concentrations of β-estradiol correlates with a decrease in yeast growth rate that is dependent on the presence of catalytically functional M^pro (*Figure 1—figure supplement 1b*). At a high expression level induced with 2 µM of β-estradiol, yeast growth rate becomes tightly coupled to M^pro function and can be used as a readout of the function of the expressed M^pro variant. While the endogenous yeast substrates are unknown, this assay is likely reporting on M^pro cleavage of numerous cellular targets. Sampling of more than one cleavage site may better represent the physiological role of M^pro, which has 11 viral and numerous host cleavage sites.

## Comprehensive deep mutational scanning of M^pro

We integrated our three screens with a systematic mutational scanning approach to determine the impact of each single amino acid change in M^pro on its function (*Figure 1d*). A comprehensive M^pro single site variant library was purchased commercially (Twist Biosciences). Each position of M^pro was mutated to all other 19 amino acids plus a stop codon, using the preferred yeast codon for each substitution. We transferred the library to a plasmid under the LexA promoter. To efficiently track each variant of the library using deep sequencing, we employed a barcoding strategy that allowed us to track mutations across the gene using a short sequence readout. We engineered the barcoded library so that each mutant was represented by 20–40 unique barcodes and used PacBio sequencing to associate barcodes with M^pro mutations (*Figure 1d*). About 96% of library variants were linked to 10 or greater barcodes (*Figure 1—figure supplement 1e*). As a control, the library was doped with a small amount of WT M^pro linked to approximately 150 barcodes.

We transformed the plasmid library of M^pro mutations into yeast strains harboring the respective reporter for each functional screen. The mutant libraries were amplified in the absence of selection and subsequently β-estradiol was added to induce M^pro expression. Variant counts analyzed by sequencing before and after the pre-selection amplification step were correlated, consistent with minimal to no selection prior to induction with β-estradiol (*Figure 1—figure supplement 1f* and *Figure 1—figure supplement 1g*). For the fluorescent screens, the cells were incubated with β-estradiol at the

concentration determined to limit M$^{pro}$ toxicity (125 nM) for the time required for WT M$^{pro}$ to achieve full reporter activity (1.5 hr for the FRET screen and 6 hr for the TF screen). Subsequently, cells were separated by FACS into populations with either uncleaved or cleaved reporter proteins (see *Figure 1a* and *Figure 1b*). For the growth screen, cells were incubated with a higher concentration of β-estradiol determined to slow yeast growth (2 µM) (*Figure 1—figure supplement 1b*). Populations of cells were collected at the 0 and 16 hr time points. For each cell population in each screen, plasmids encoding the mutated M$^{pro}$ library were recovered, and the barcoded region was sequenced using single end Illumina sequencing. For the TF and FRET screens, the functional score of each mutant was calculated as the fraction of the mutant in the cut population relative to its fraction in both populations. For the growth screen, the functional score was calculated as the fraction of the mutant at the 0 hr time point relative to the fraction in the 0 hr and 16 hr time points. We normalized the functional scores in all three screens to facilitate comparisons, setting the score for the average WT M$^{pro}$ barcode as 1 and the average stop codon as 0 (see *Figure 2—source data 1* for all functional scores).

To analyze the reproducibility of each screen, we performed biological replicates. For each biological replicate we separately transformed the library into yeast cells, and independently performed competition experiments and sequencing. Functional scores between replicates were strongly correlated ($R^2$>0.98 for all three screens, *Figure 2a*) and we could clearly distinguish between functional scores for WT M$^{pro}$ and those containing stop codons (*Figure 2b*). There was a narrow distribution of functional scores for stop codons in all the three screens across the M$^{pro}$ sequence except at the last seven positions (amino acids 300–306) (*Figure 2c*), supporting previous experiments showing that these residues are dispensable for M$^{pro}$ activity and the importance of residue Q299 for M$^{pro}$ function (*Lin et al., 2008*). We categorized functional scores as WT-like, intermediate, or null-like based on the distribution of WT barcodes and stop codons in each screen (*Figure 2d* and *Figure 2—figure supplement 1*). Heatmap representations of the functional scores determined in replicate 1 of all three screens are shown in *Figure 3* (FRET screen), *Figure 3—figure supplement 1* (TF screen), and *Figure 3—figure supplement 2* (growth screen).

## Comparison between three screens

Comparing the average functional score at each position (a measure of mutational sensitivity) between the three screens shows a strong correlation (*Figure 4a–c*). The principal differences lie in the sensitivity of the screens to mutation, with the average defective mutation in the growth screen being more exaggerated than that in the fluorescent-based screens (*Figure 4c*). The scores in the growth screen are likely integrating cutting efficiency over a diverse set of cleavages sites which may contribute to this screen's increased sensitivity to mutation. Despite these differences, there are striking correlations in the mutational patterns of M$^{pro}$ across all three screens as can be visualized in the heatmap of average scores per position and when mapped to M$^{pro}$'s structure (*Figure 4a and b*). These similarities indicate that the three screens are reporting the same fundamental biophysical and biochemical constraints of the protein.

Several lines of evidence indicate that the functional scores are biochemically and biologically relevant. First, we compared the scores to previously published studies of point mutations (*Figure 4d* and *Figure 4—source data 2*). For example, mutating the residues of the catalytic dyad, C145 and H41, inactivates the protease both in our screen and in in vitro biochemical assays as expected (*Hegyi et al., 2002*). Additionally, in vitro assays have shown that residues at the dimer interface including S10, G11, and E14 are essential for SARS-CoV-1 M$^{pro}$ dimerization and function (*Chen et al., 2008*). Mutations at these residues are also deleterious to M$^{pro}$ function in our screen. Because of the high sequence and functional similarities between SARS-CoV-1 and CoV-2 M$^{pro}$, we expect that the majority of the mutational analyses performed previously on SARS-CoV-1 M$^{pro}$ will be valid for SARS-CoV-2 M$^{pro}$. We examined how the dynamic range of our screens relate to catalytic measurements. The growth screen measurements exhibited a linear pattern with relative catalytic rates previously reported for individual variants (*Figure 4d*). In contrast, the TF screen results showed a non-linear pattern, reminiscent of a binding equation. To assess these patterns in a systematic manner, we fit the graphs to both a linear equation and a non-linear binding equation with initial parameters of 1:1 for the linear fit, and an inflection point of 0.5 for the non-linear equation. Using this approach, we observed an apparent non-linear relationship between the functional scores measured in both the FRET and TF screens, and the relative catalytic activity of mutants measured independently for M$^{pro}$ in vitro in various studies

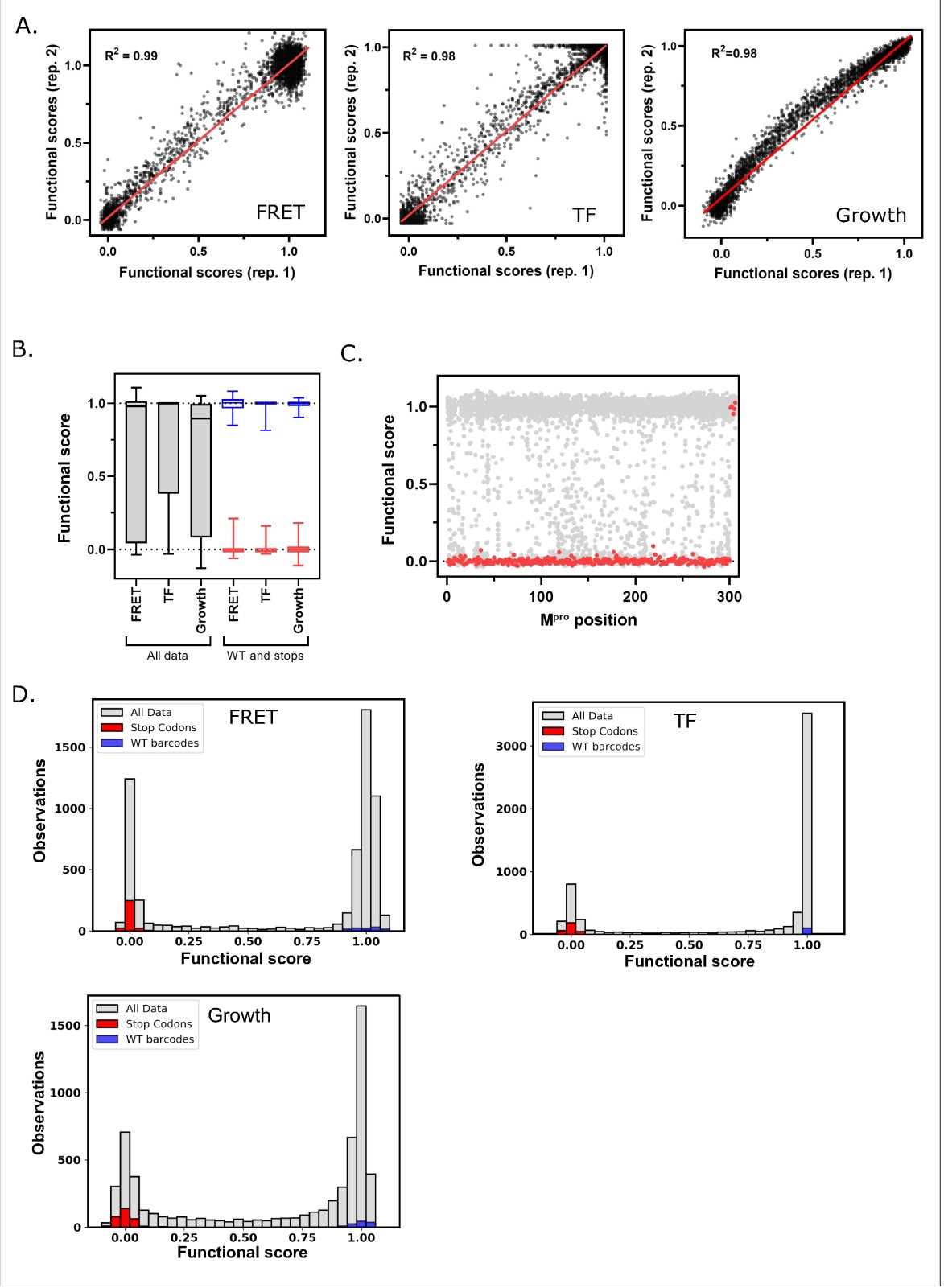

**Figure 2.** Main protease (M^pro) functional scores are reproducible, and variants can be clearly distinguished based on function. (**A**) Correlation between biological replicates of functional scores of all M^pro variants for each screen. Red line indicates best fit. (**B**) Distribution of functional scores for all variants (gray), stop codons (red), and wild-type (WT) barcodes (blue) in each screen. (**C**) The functional scores for all variants (gray) and stop codons (red) at each position of M^pro in the fluorescence resonance energy transfer (FRET) screen. (**D**) Distribution of all functional scores (gray) in each screen. Functional

*Figure 2 continued on next page*

*Figure 2 continued*

scores are categorized as WT-like, intermediate, or null based on the distribution of WT barcodes (blue) and stop codons (red) in each screen. See *Figure 2—source data 1*.

The online version of this article includes the following source data and figure supplement(s) for figure 2:

**Source data 1.** Sequencing counts and functional scores for each amino acid of main protease (M^pro) in both replicates of all three screens.

**Figure supplement 1.** Cumulative frequency distributions for all variants (gray), stops (red), and wild-type (WT) barcodes (blue) for all three screens.

($R^2$=0.81 for non-linear fit to TF screen and $R^2$=0.93 for non-linear fit to FRET screen) (*Figure 4d*). Compared to the fluorescent screens, there is a stronger linear relationship ($R^2$=0.86) between the scores measured in our growth screen and the catalytic efficiencies of the individual mutants. These analyses indicate that the growth screen more fully captures the dynamic range of mutations with small functional defects that tend to appear WT-like in the FRET and TF screens. For the remainder of this paper, we will report the functional scores collected for the FRET and growth screens in the main figures, and the TF screen in the supplementary figures. The advantage of the functional scores for each mutant from the FRET screen is that they report direct cleavage of a defined substrate, with the drawback being that they exhibit less sensitivity to mutation. The advantage of the growth screen is that the functional scores show a more linear relationship with catalytic rate, while the drawback is that the screen reports cleavage of undefined substrates. Because of the correlation between all three screens, similar overall biophysical conclusions are supported by each screen.

## Functional characterization of natural M^pro variants

To further assess the scores from our screens, we examined the functional scores of the M^pro variants observed in clinical samples. Because M^pro is essential for viral replication, deleterious mutations should be purged from the circulating population. The CoV-Glue-Viz database archives all mutations observed in the GISAID human SARS-CoV-2 sequences sampled from the ongoing COVID-19 pandemic (*Singer et al., 2020*). We compared the frequency at which the clinical variants of the M^pro gene (ORF1ab/nsp5A-B) have been observed to their functional scores. The vast majority of the clinical isolates that have been sequenced to date have either 0 or 1 M^pro mutations with fewer than 0.4% having 2 or more mutations and thus we did not account for epistasis in our analysis. We found that the most abundant clinical variants are highly functional in our assays (*Figure 5a* [FRET and growth screens] and *Figure 5—figure supplement 1a* [TF screen]); however, lower frequency variants in clinical samples were found to have a wide range of M^pro function. Surprisingly, M^pro sequences among the clinical samples include premature stop codons that have been observed up to 100 times (out of over 5 million total isolates to date) (*Figure 5a* [FRET and growth screens] and *Figure 5—figure supplement 1a* [TF screen]). Because M^pro function is required for viral fitness, we assume that the frequency of stop codons observed in the data is an indication of sequencing error in the clinical samples. Accounting for this sequencing error, we examined the functional score of the 290 non-synonymous mutations in the M^pro gene that have been observed more than 100 times. The vast majority of these clinical variants exhibit WT-like function with only nine having a score below that of the WT distribution (see *Figure 5a–c*). This observed enrichment for variants with WT-like function in the circulating SARS-CoV-2 virus indicates that M^pro is undergoing strong purifying selection in the human population.

Additionally, we examined the experimental function of M^pro mutations compared with the diversity of M^pro in viruses related to SARS-CoV-2. There is a 96% sequence identity between the SARS-CoV-2 and the SARS-CoV-1 M^pro proteases, with only 12 amino acid differences. In our study, all of the amino acid differences in SARS-CoV-1 M^pro are WT-like in SARS-CoV-2, underscoring the credibility of the functional scores (*Figure 5b* [FRET and growth screens] and *Figure 5—figure supplement 1b* [TF screen]). We went on to analyze the diversity in 852 sequences across a set of M^pro homologs with an average homology of 47% from genetically diverse coronaviruses. We identified 1207 amino acid changes located at 263 positions of M^pro and examined the functional score of these variants in our data. Here again, we saw enrichment toward functional M^pro variants with only 6% (77 out of 1207) natural variants having functional scores in the FRET screen below the WT range (*Figure 5b* and *Figure 5c* [FRET and growth screens] and *Figure 5—figure supplement 1b* [TF screen]). Further analysis of these deleterious variants should provide insight into the role epistasis played in the historical

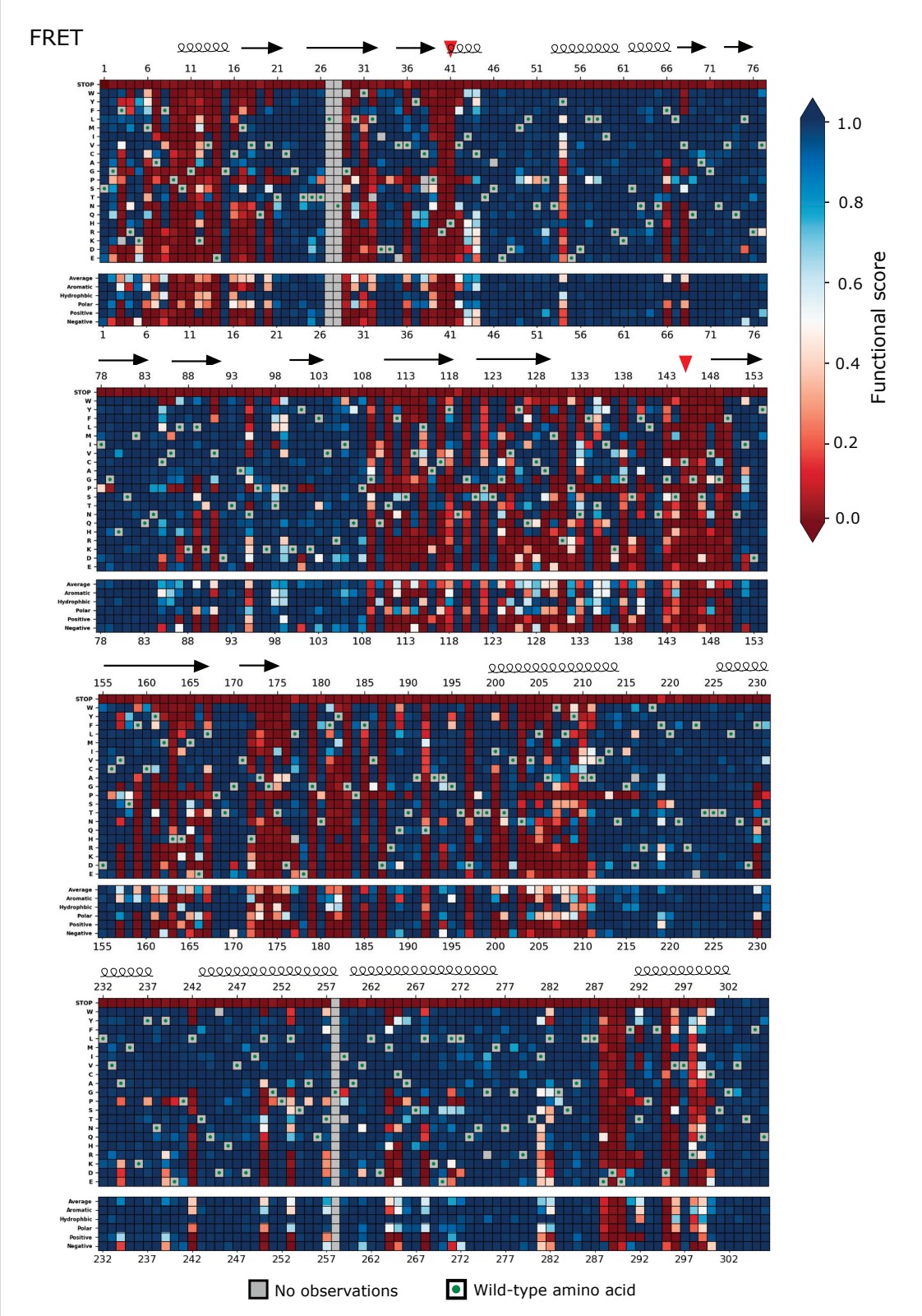

**Figure 3.** Heatmap representation of the main protease (M^pro) functional scores measured in the fluorescence resonance energy transfer (FRET) screen (replicate 1). Arrows represent positions that form β-sheets, coils represent α-helices, and red triangles indicate the catalytic dyad residues H41 and C145.

*Figure 3 continued on next page*

*Figure 3 continued*

The online version of this article includes the following figure supplement(s) for figure 3:

**Figure supplement 1.** Heatmap representation of scores from the transcription factor (TF) screen (replicate 1).

**Figure supplement 2.** Heatmap representation of scores from the growth screen (replicate 1).

evolution of M^pro, and these insights may have utility in the generation of future pan-coronavirus inhibitors.

## Structural distribution of mutationally sensitive M^pro positions

Invariant sites that are essential to M^pro function are promising targets for designing inhibitors. About 24 positions of M^pro exhibited low mutation tolerance, defined as 17 or more substitutions with null-like function: P9, S10, G11, E14, R40, H41, T111, S113, R131, C145, G146, S147, G149, F150, H163, G174, G179, G183, D187, D197, N203, D289, E290, and D295 (*Figure 6a*). Only four of these mutation-sensitive residues contact the substrate: H41 and C145 (the catalytic residues), as well as H163 and D187. H163 interacts with the invariable P1 Gln of the substrate and D187 forms a hydrogen bond with a catalytic water and a salt bridge with R40. A large body of work has previously shown that dimerization is indispensable to M^pro function (*Chou et al., 2004*; *Hsu et al., 2005a*, *Chen et al., 2008*; *Cheng et al., 2010*). Our study also supports the critical functional role of dimerization as we see prevalent mutation-sensitivity in residues at the dimer interface, including P9, S10, G11, E14, and E290, each of which cannot be altered without complete loss of function.

Outside of these well-studied critical M^pro sites, there are additional clusters of mutation-intolerant residues. The R131, D197, N203, D289, and E290 lie at the interface of Domain II and Domain III sandwiched between dimers and make up part of a surface identified by structural modeling as a possible distal drug binding pocket (*Bhat et al., 2021*; *Weng et al., 2021*; *Figure 6b*). Within this cluster, a dynamic salt bridge is formed between R131 located on the loop of Domain II connecting β10–11 of the catalytic pocket, and D289 in the α-helical Domain III that has been reported to contribute to the flexibility and structural plasticity of M^pro (*Bhat et al., 2021*). The location of these residues at the interface of the two domains and the dimer interface, combined with the fact that they are critical to M^pro function suggests that they are part of a distal regulatory communication network. Our studies clearly indicate the critical function played by this network of residues providing motivation for further examination of their potential as a mutation-resistant target for inhibitor design.

A second cluster of mutation-intolerant residues appear to be part of an allosteric communication network between the active site and the dimerization interface. Prior studies of individual mutations also suggest allosteric connections between the dimerization and active sites. Mutations at both E166 (*Cheng et al., 2010*) and S147 (*Barrila et al., 2006*) were found to disrupt dimerization. Both positions E166 and S147 are located distal to the dimerization site, suggesting that the properties of these two sites are interdependent. Our results show that there is a physically interacting chain of mutation-sensitive residues that bridge from the active site to the dimerization site (*Figure 6c*). This bridge is composed of H163 that directly contacts the P1 Gln of substrate, S147, L115, and S10 at the dimer interface. Each of these dimer-to-active site bridging residues are critical to M^pro function and are strongly conserved among M^pro homologs. Based on these observations, we suggest that the physical interactions between H163, S147, L115, and S10 mediate critical communication between the active sites of both subunits in the M^pro dimer.

All 24 of the identified mutation-intolerant residues are highly conserved among SARS-CoV-2 M^pro homologs (*Figure 6d* [FRET and growth screens] and *Figure 6—figure supplement 1* [TF screen]). While functional hot spots accurately predict evolutionary conservation, conservation does not accurately predict functional hot spots. There are many residues in M^pro that are strongly conserved, but that can be mutated without strong impacts on function. This pattern has been widely observed for other proteins (*Hietpas et al., 2011*; *Melamed et al., 2013*; *Roscoe et al., 2013*; *Starita et al., 2013*; *Mishra et al., 2016*). While many features distinguish natural evolution and experimental studies of fitness (*Boucher et al., 2019*) one of the outstanding differences is the strength of selection. While functional hot spots can be defined by strong impacts on function that are experimentally measurable, small fitness changes that may be too small for experimental resolution can drive selection in natural evolution due to large population sizes and timescales (*Ohta, 1973*). Our functional screen

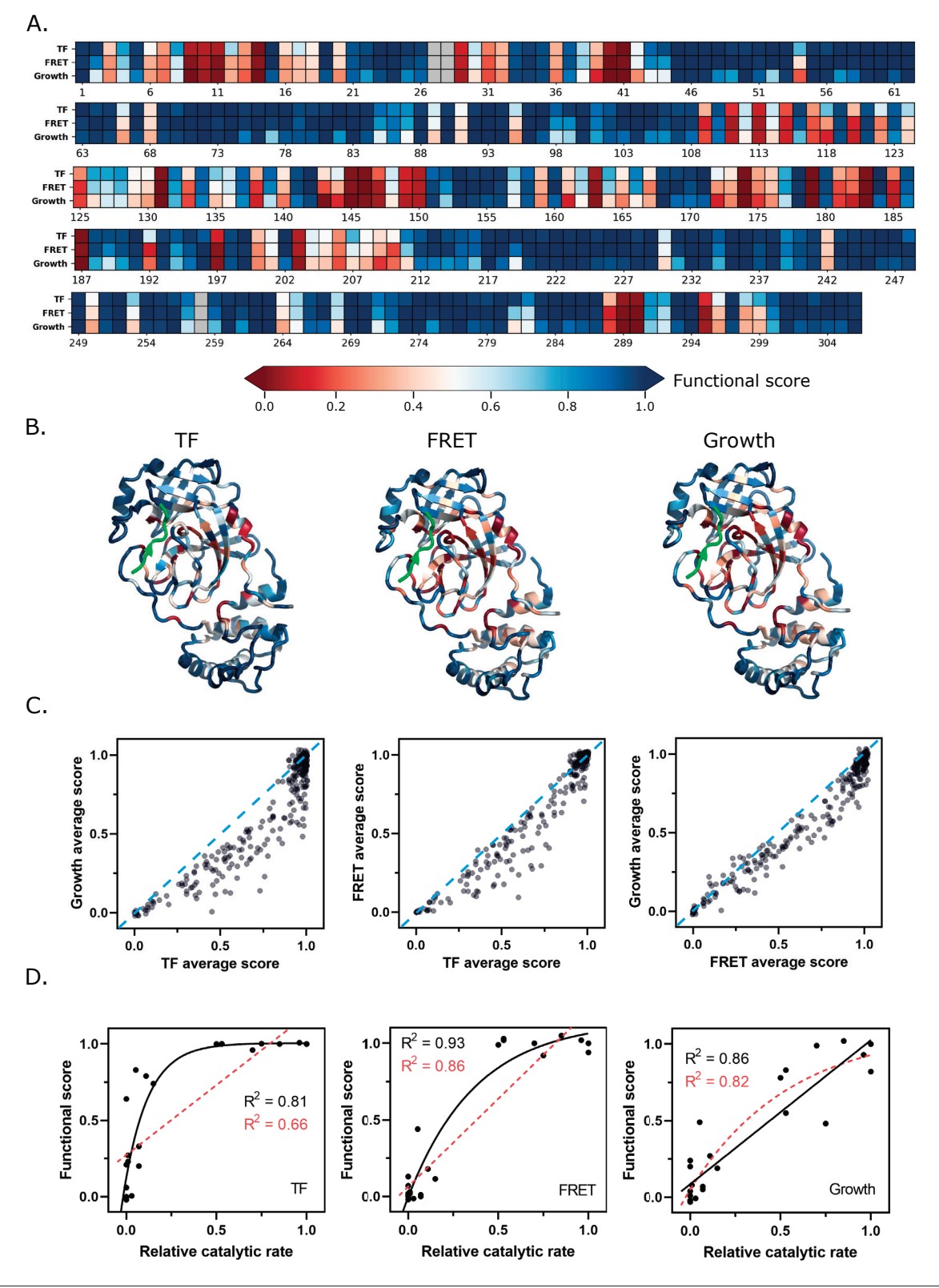

**Figure 4.** Functional scores reflect fundamental biophysical constraints of main protease (M$^{pro}$). (**A**) Heatmap representation of the average functional score at each position (excluding stops) in replicate 1 of each screen (see **Figure 4—source data 1**). (**B**) The average functional score at each position mapped to M$^{pro}$ structure for each screen. The Nsp4/5 substrate peptide is shown in green (PDB 7T70). (**C**) The average functional score at each position compared between the three screens. The diagonal is indicated with a blue dashed line. (**D**) Comparison between relative catalytic rates measured

*Figure 4 continued on next page*

*Figure 4 continued*

independently in various studies and functional scores measured in each screen (see *Figure 4—source data 2*). Each graph is fit with a non-linear and linear regression with the best of the two fits represented with a black solid line and the worst fit represented with a red dashed line. The non-linear regression is fit to the equation $Y = Y_m - (Y_0 - Y_m) e^{-kx}$.

The online version of this article includes the following source data for figure 4:

**Source data 1.** Average functional score (excluding stops) at each position of main protease (M^pro) in replicate 1 of each screen.

**Source data 2.** Comparison of previously measured relative catalytic rates of individual mutations to functional scores.

captures the mutations that are critical to catalytic function while evolutionary conservation depicts a wide range of mutations including those that make more nuanced contributions to function. When designing drugs to disrupt M^pro function, we hypothesize that it will be important to focus on the functionally critical sites which are a subset of the evolutionarily conserved positions.

## Functional variability at key substrate and inhibitor-contact positions

M^pro function is essential for SARS-CoV-2 replication, making it a key drug target. To help further guide inhibitor design, we assessed the mutations that are compatible with function and that should be readily available to the evolution of drug resistance. We focused these analyses on the active site, which is the target binding site for most inhibitors that have been generated against M^pro (*Cho et al., 2021*). In *Figure 7a* and *Figure 7—figure supplement 1a*, we highlight all the M^pro residues that contact the Nsp4/5 peptide, either through hydrogen bonds or van der Waals interactions (*Shaqra et al., 2022*). In our functional screens, we found dramatic variability in mutational sensitivity at these substrate-contact positions. For example, residues G143, H163, D187, and Q192 were extremely sensitive to mutation, while residues M49, N142, E166, and Q189 were highly tolerant. Despite the diverse sequence variation amongst M^pro's substrates, they occupy a conserved volume in the active site, known as the substrate envelope, and the interactions between M^pro's residues and all of its substrates are highly conserved (*Shaqra et al., 2022*) indicating that our mutation results from the Nsp4/5 cut-site will likely translate to other cut-sites.

Even among residues whose side chains make direct hydrogen bonds with substrates are positions that are surprisingly tolerant to mutation, namely N142, E166, and Q189. N142 forms distinct hydrogen bonds with Nsp4/5 and Nsp8/9, which has been proposed as a mechanism of M^pro substrate recognition (*MacDonald et al., 2021*). Q189 is in a flexible loop that closes over the substrates, allowing accommodation of diverse cut-sites (*Shaqra et al., 2022*). In our screens, we find that these proposed substrate-recognition positions are very tolerant to mutation (*Figure 7b* [FRET and growth screens] and *Figure 7—figure supplement 1b* [TF screen]) and have high potential for developing inhibitor resistance. Our results indicate that mutations at N142, E166, and Q189 are compatible with function and are readily available to the evolution of drug resistance.

A recent study comprehensively examined 233 X-ray crystal structures of SARS-CoV-2 M^pro in complex with a wide range of inhibitors (*Cho et al., 2021*). In 185 of these 233 structures, inhibitors lie in the same binding pocket in the active site, primarily contacting M^pro positions T25, H41, M49, N142, S144, C145, H163, H164, E166, P168, H172, Q189, and A191. We therefore went on to determine the mutations at these key inhibitor binding residues that are compatible with M^pro function and should likely be available to resistance evolution. *Figure 7c* and *Figure 7—figure supplement 1c* illustrate a representative structure of M^pro bound to the N3 inhibitor with the average mutational sensitivity of each position mapped to the structure by color (*Jin et al., 2020*). In addition, heatmaps are shown detailing the mutations at these positions that are compatible with function (*Figure 7—figure supplement 1d*). Of note, residues N142, E166, and Q189 form direct hydrogen bonds with many M^pro inhibitors and most mutations at these positions result in a functional protease. Additionally, T25, M49, M164, P168, and A191 form van der Waals interactions with a variety of inhibitors suggesting that mutations at these positions could disrupt inhibitor interactions while maintaining M^pro function. In contrast, positions H41, S144, C145, H163, and H172 are highly sensitive in our screen, as well as strongly conserved in nature, and therefore would be ideal contact positions for inhibitors with reduced likelihood of evolving M^pro resistance.

Pfizer has developed the first FDA-authorized M^pro inhibitor, PF-07321332 (*Owen et al., 2021*). We examined the structure of M^pro bound to PF-07321332 to identify positions with the potential to evolve

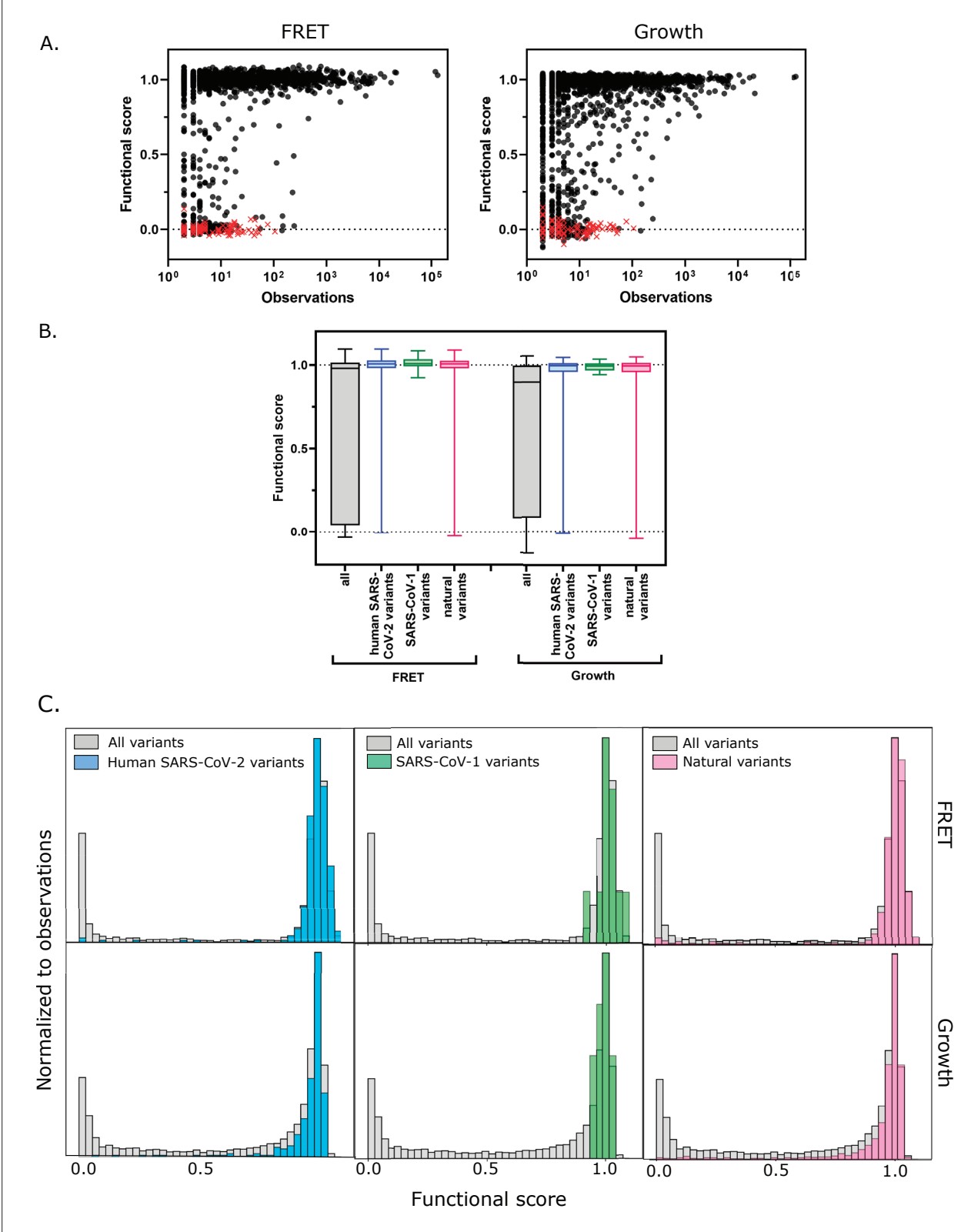

**Figure 5.** Functional scores indicate that natural amino acid variants of main protease (Mᵖʳᵒ) are generally fit. (**A**) Comparison of functional scores in the FRET screen (left panel) and growth screen (right panel) to the number of observations among clinical samples. All missense mutations excluding stops are indicated with black circles and stop codons are indicated with red x's. (See *Figure 5—source data 1*) (**B**) The distribution of functional scores of all variants in the FRET and growth screens compared to the observed clinically-relevant Mᵖʳᵒ variants (human SARS-CoV-2 variants, blue), 12 amino

*Figure 5 continued on next page*

*Figure 5 continued*

acid differences between SARS-CoV-2 and SARS-CoV-1 (green), and the different amino acids in a broad sample of Mpro SARS-CoV-2 homologs (natural variants, pink). Distributions are significantly different as measured by a two-sample Kolmogorov-Smirnov (KS) (All FRET vs. human SARS-CoV-2 variants: N = 6044, 289, p<0.0001, D = 0.3258; All FRET vs. SARS-CoV-1 variants: N=6044, 12, p=0.0398, D=0.4223; All FRET vs. natural variants: N = 6044, 1205, p<0.0001, D = 0.2984; All Growth vs. human SARS-CoV-2 variants: N = 6044, 289, p<0.0001, D = 0.3938; All growth vs. SARS-CoV-1 variants: N=6044, 12, p=0.0024, D=0.5533; All growth vs. natural variants: N=6044,1205, p<0.0001, D = 0.3462) (**C**) Histogram of functional scores of all variants (grey) compared to that of human SARS-CoV-2 variants (blue), SARS-CoV-1 variants (green), and natural variants (pink).

The online version of this article includes the following source data and figure supplement(s) for figure 5:

**Source data 1.** Frequency at which the clinical variants of the main protease (Mpro) gene have been observed.

**Figure supplement 1.** Functional scores indicate that natural amino acid variants of main protease (Mpro) are generally fit.

resistance against this drug (*Figure 7d* [FRET and growth screens] and *Figure 7—figure supplement 1e* [TF screen]; *Zhao et al., 2021*). Evolutionarily accessible resistance mutations are single base change mutations that would disrupt inhibitor binding while maintaining WT-like substrate recognition and cleavage. We identified all mutations of Mpro that have WT-like function in both the FRET and growth screens, would lead to a predicted decrease in inhibitor binding energy upon mutation of greater than 1 kcal/mol, and are accessible with a single nucleotide base change. These criteria led to the identification of three mutations, Q189E, E166A, and E166Q with potential resistance against PF-07321332. These three positions are at sites where the inhibitor protrudes out of the defined substrate envelope, providing further evidence that these residues may evolve inhibitor resistance while maintaining substrate recognition (*Shaqra et al., 2022*). Of note, Q189E is a natural variant in both the avian infectious bronchitis virus and the swine coronavirus, HKU15 CoV, widely detected in pigs in Asia and North America and of pandemic concern due to its ability to replicate in human cells (*Edwards et al., 2020*). PF-07321332 may have reduced efficacy against these concerning homologs due to its decreased interactions with Q189E Mpro.

In addition to the impacts on side-chain properties, mutations in Mpro may also impact resistance through changes in main-chain conformation and dynamics, particularly in loops. In-depth structural analyses will be important to extensively assess the potential impacts of mutations on resistance through these mechanisms. Of note, mutations at N142 appears of particular interest for further investigation of conformational changes that may impact resistance evolution. N142 is mutation tolerant and located in a loop over the P1 position of the substrate. The lactam ring on PF-07321332 protrudes outside of the substrate envelope at this location (*Shaqra et al., 2022*). Mutations at position 142 should be readily available to Mpro evolution and appear likely to influence loop conformation at a site where PF-07321332 extends beyond the substrate envelope. Together these observations suggest that N142 warrants further attention as a potential contributor to drug resistance.

## Discussion

During the SARS-CoV-2 pandemic, intensive efforts have been launched to rapidly develop vaccines and anti-viral drugs to improve human health. In this study, we provide comprehensive functional information on a promising therapeutic target, Mpro, with the hopes that these results will be useful in the design of more effective and long-lasting anti-SARS-CoV-2 drugs. We built three yeast screens to measure the functional effects of all individual amino acid changes in Mpro. The resulting fitness landscapes provide information on residues to both target and avoid in the drug design process. In the active site, the primary current target of Mpro inhibitors, our results indicate both mutation-sensitive positions that provide ideal anchors for inhibitors and mutation-tolerant positions to avoid. Among the positions to avoid, Q189 is noteworthy because it forms hydrogen bonds directly with substrates (*MacDonald et al., 2021*; *Shaqra et al., 2022*), contacts promising Mpro drugs such as PF-07321332 (*Cho et al., 2021*; *Owen et al., 2021*; *Zhao et al., 2021*), is a natural variant in coronaviruses of future pandemic concern, and is surprisingly tolerant of mutations in our screen.

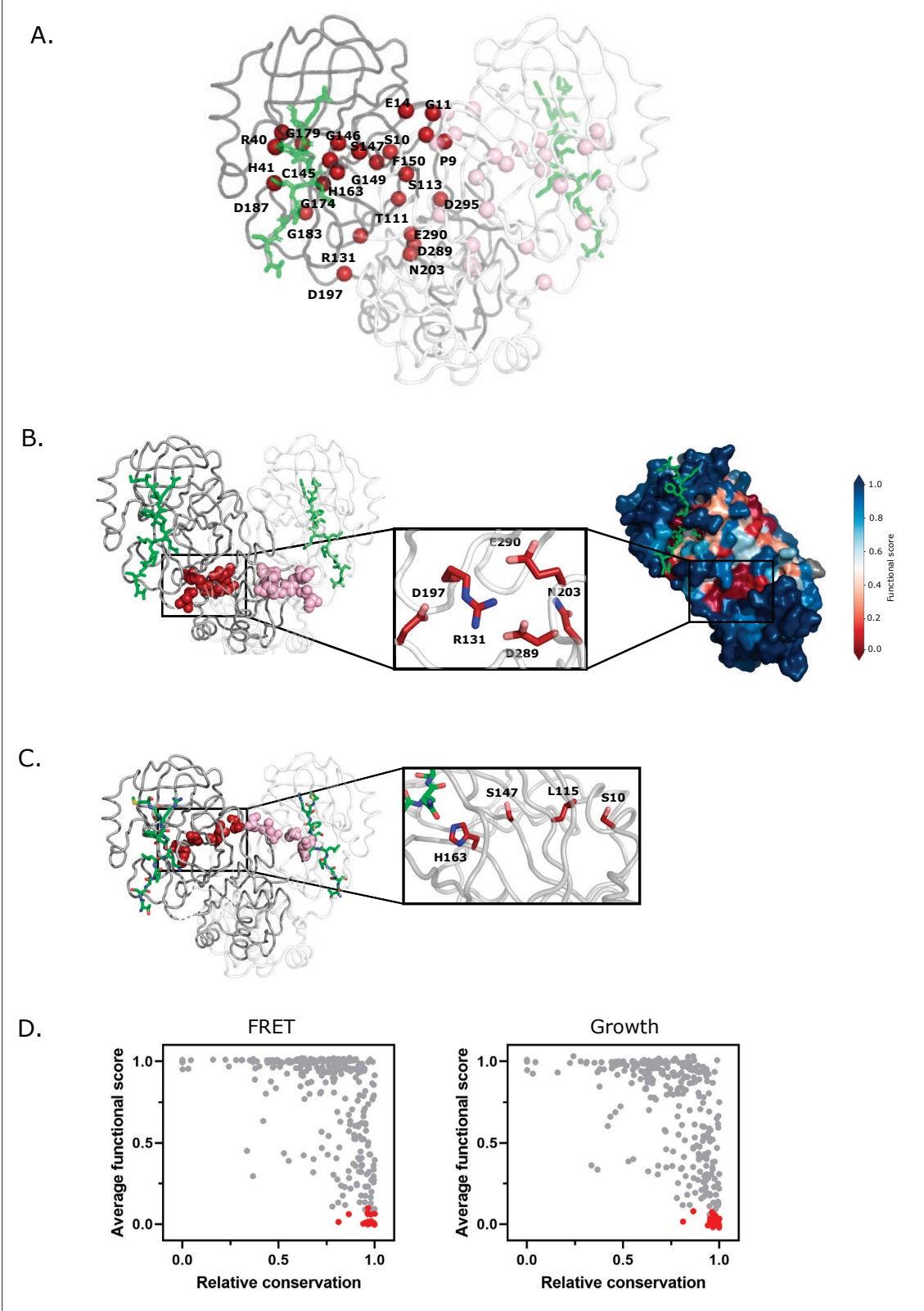

**Figure 6.** Structural distribution of main protease (M^pro) positions that are intolerant to mutation. (**A**) M^pro positions that are intolerant of mutations with 17 or more substitutions having null-like function are represented by red spheres on chain A (shown in gray) and pink spheres on chain B (shown in white). The Nsp4/5 substrate peptide is shown in green (PDB 7T70). (**B**) Representation of a cluster of the mutation-intolerant positions (red spheres) at a site distal to the active site. (**C**) A cluster of mutation-intolerant residues (red spheres) appear to be part of a communication network between the active

*Figure 6 continued on next page*

*Figure 6 continued*

site and the dimerization interface. (**D**) Comparison of the average functional score of each position to conservation observed in a broad sample of severe acute respiratory syndrome coronavirus-2 (SARS-CoV-2) M$^{pro}$ homologs. The 24 mutation-intolerant positions shown as red spheres in part A are highlighted in red. Positions exhibiting the strongest evolutionary conservation exhibit a broad range of experimental sensitivity to mutation while the most evolutionary variable positions are experimentally tolerant to mutations.

The online version of this article includes the following figure supplement(s) for figure 6:

**Figure supplement 1.** Comparison of the average transcription factor (TF) functional score of each position to conservation observed in a broad sample of severe acute respiratory syndrome coronavirus-2 (SARS-CoV-2) main protease (M$^{pro}$) homologs.

We found that the functional scores measured from all three distinct screens were highly correlated, that they identified known critical M$^{pro}$ residues, and that clinical variants were overwhelmingly functional, indicating that the scores successfully capture key biochemical and functional properties of M$^{pro}$. However, there are a couple of caveats that should be kept in mind when utilizing these datasets. For example, we do not fully understand how M$^{pro}$'s biochemical function relates to viral fitness. Having some M$^{pro}$ function is essential to the virus, so mutations that destroy M$^{pro}$ function will form non-functional viruses. Function-fitness relationships tend to be non-linear (*Heinrich and Rapoport, 1974*; *Kacser and Fell, 1995*; *Jiang et al., 2013*) and it may be likely that M$^{pro}$ function must be decreased by a large amount in order to cause measurable changes in viral replication efficiency. This relationship between M$^{pro}$ function and SARS-CoV-2 fitness would need to be determined in order to translate our functional scores to fitness scores. Additionally, our TF and FRET screens quantify cleavage at one defined site (Nsp4/5) and it may be important to analyze all sites in order to fully understand the selection pressures acting on M$^{pro}$. Another important caveat is that our fitness landscape captures single amino acid changes and therefore does not provide information on the potential interdependence or epistasis between double and higher order mutations. Information regarding epistasis will be important for accurately predicting the impacts of multiple mutations on fitness. Despite these caveats, the similarity in fitness landscapes for the TF and FRET screens with the yeast growth screen suggests that all three capture fundamental and general aspects of M$^{pro}$ selection. In addition, the high function of almost all naturally occurring substitutions in the diversity of natural M$^{pro}$ sequences indicates that estimates of fitness effects in different genetic backgrounds can be made based on our results.

We believe that our results will be a useful guide for the continuing intense efforts to develop drugs that target M$^{pro}$ and the interpretation of future M$^{pro}$ evolution in the face of drug pressure. In particular, our results identify amino acid changes that can be functionally tolerated by M$^{pro}$ that are likely to disrupt binding to inhibitors. In a recent study, Shaqra, Schiffer and colleagues mapped the M$^{pro}$ substrate envelope; locations where the inhibitors protrude from this envelope is an indicator of susceptibility to resistance mutations (*Shaqra et al., 2022*). The information in these two studies provides a new view into resistance evolution that can be incorporated into ongoing drug design efforts. Locations in the active site as well as at a likely allosteric site that cannot readily evolve without compromising function are ideal targets for anchoring inhibitors with reduced potential to evolve drug resistance.

Our next steps involve developing efficient strategies for assaying M$^{pro}$ fitness landscapes in the presence of potential inhibitors in order to define structure-resistance relationships. This would provide critical guidance for reducing the likelihood of resistance at earlier stages of drug development than is currently possible. For example, it would identify inhibitors with the least likelihood of developing resistance. It would also provide the potential for identifying inhibitors with non-overlapping resistance profiles that if used in combination would not be susceptible to resistance from an individual mutation. There are technical hurdles to overcome in using our yeast-based screens to investigate resistance because many small molecules are ineffective due to poor permeability and/or export from yeast. We are assessing strategies to both increase the druggability of yeast and porting our assays to mammalian cells (*Chinen et al., 2017*). The results from our current work on M$^{pro}$ in yeast as well

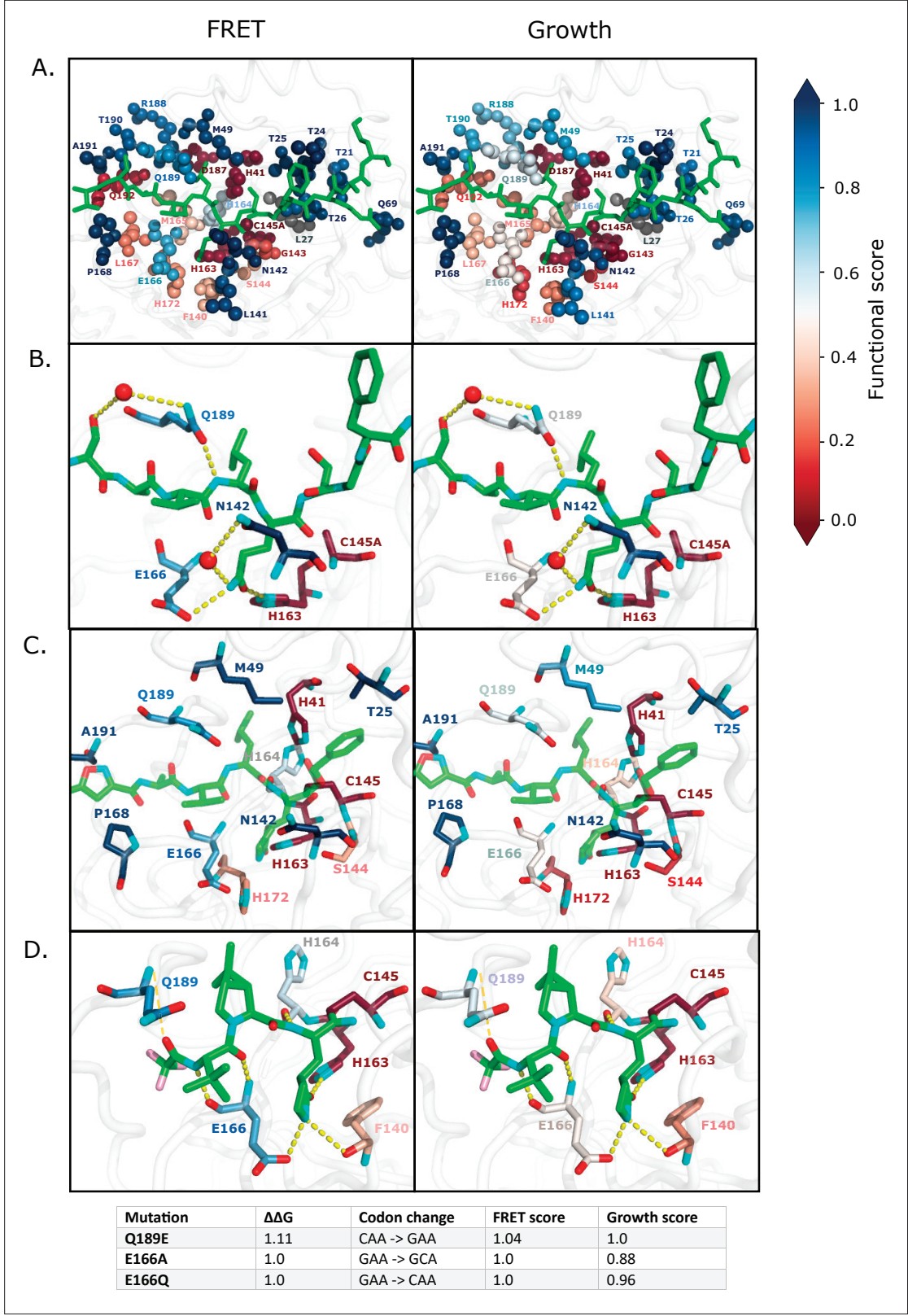

| Mutation | ΔΔG | Codon change | FRET score | Growth score |
|----------|------|--------------|------------|--------------|
| Q189E | 1.11 | CAA -> GAA | 1.04 | 1.0 |
| E166A | 1.0 | GAA -> GCA | 1.0 | 0.88 |
| E166Q | 1.0 | GAA -> CAA | 1.0 | 0.96 |

**Figure 7.** Substrate and inhibitor binding sites are variably sensitive to mutation. (**A**) All main protease (M^pro) positions that contact the Nsp4/5 substrate peptide are represented in spheres and colored by their average fluorescence resonance energy transfer (FRET) functional score (left panel) and growth functional score (right panel; PDB 7T70). The Nsp4/5 peptide is shown in green. (**B**) M^pro positions that form hydrogen bonds with the Nsp4/5 substrate are shown in sticks and colored by their average FRET functional score (left panel) and growth functional score (right panel; PDB 7T70). Oxygens are

*Figure 7 continued on next page*

*Figure 7 continued*

shown in red and nitrogens in cyan. Water molecules are represented as red spheres and hydrogen bonds as yellow dashed lines. (**C**) M^pro positions shown to contact over 185 inhibitors in crystal structures (*Cho et al., 2021*) are shown in sticks and are colored by their average FRET functional score (left panel) and average growth functional score (right panel). Shown is a representative structure of M^pro bound to the N3 inhibitor (PDB 6LU7) (*Jin et al., 2020*). The N3 inhibitor is shown in green, oxygens in red, and nitrogens in cyan. (**D**) M^pro positions that form hydrogen bonds with the Pfizer inhibitor, PF-07321332, are represented by sticks and colored by their average FRET functional score (left panel) or growth functional score (right panel; PDB 7VH8) (*Owen et al., 2021*; *Zhao et al., 2021*). PF-07321332 is shown in green, oxygens in red, nitrogens in cyan, fluorines in pink. Hydrogen bonds less than 4 Å are represented with thick yellow dashed lines and greater than 4 Å with a thin yellow dashed line. The table below lists the mutations with highest potential for being resistant against PF-07321332.

The online version of this article includes the following figure supplement(s) for figure 7:

**Figure supplement 1.** Substrate and inhibitor binding sites are variably sensitive to mutation.

as previous studies using fitness landscapes to analyze drug resistance in other proteins (*Deng et al., 2012*; *Choi et al., 2014*; *Firnberg et al., 2014*; *Ma et al., 2017*) indicates a strong potential of these approaches to improve our understanding and ability to combat resistance evolution.

# Materials and methods

**Key resources table**

| Reagent type (species) or resource | Designation | Source or reference | Identifiers | Additional information |
|---|---|---|---|---|
| Gene (SARS-CoV-2) | ORF1ab/ nsp5A-B | NIH GenBank | NC_045512 | M^pro |
| Strain, Strain background (*Saccharomyces cerevisiae*) | W303 | *Saccharomyces* Genome Database | GenBank JRIU00000000 | |
| Antibody | anti-his tag HRP-labelled (Mouse monoclonal) | R&D systems | CAT#: MAB050H | WB (1:4000) |
| Recombinant DNA reagent | Barcoded UbM^pro plasmid library | This paper | p416LexA-UbM^pro(lib)-N18 | See Materials and Methods section "Generating mutant libraries" |
| Recombinant DNA reagent | Barcoded WT UbM^pro plasmid | This paper | p416LexA-UbM^pro(WT)-N18 | See Materials and Methods section "Construction of WT Ub-M^pro vector" |
| Recombinant DNA reagent | C145A-M^pro-his$_6$ plasmid | This paper | p416LexA-UbM^pro(C145A)-his | See Materials and Methods section "Analysis of M^pro expression" |
| Recombinant DNA reagent | pCyPet-His | Addgene | #14,040 | |
| Recombinant DNA reagent | pYPet-His | Addgene | #14,031 | |
| Recombinant DNA reagent | CyPet-MproCS-YPet fusion gene | This paper | | See Materials and Methods section "Generating FRET strain" |
| Recombinant DNA reagent | pDK-ATC | PMID:28660202 | | Integrative bidirectional plasmid with TEF and CUP promoters |
| Recombinant DNA reagent | pDK-ATG | PMID:28660202 | | Integrative bidirectional plasmid with TEF and GPD promoters |
| Recombinant DNA reagent | DBD-M^proCS-AD fusion gene | This paper | | See Materials and Methods section "Generating split TF strain" |
| Commercial assay or kit | KAPA SYBR FAST qPCR Master Mix | Kapa Biosystems | KK4600 | |

*Continued on next page*

*Continued*

| Reagent type (species) or resource | Designation | Source or reference | Identifiers | Additional information |
|---|---|---|---|---|
| Commercial assay or kit | BCA protein assay kit | Pierce | CAT# 23,225 | |
| Chemical compound, drug | β-Estradiol | Sigma Aldrich | E2768 | |
| Software, algorithm | Scripts to tabulate variant counts | This paper | https://github.com/Julia Flynn/BolonLab, (copy archived at swh:1:rev:b54d80818 c2681fb89533ae330c 18a3d39f32ab6) | See Materials and Methods section "Analysis of Illumina sequencing data" |
| Software, algorithm | Scripts to associate barcodes with variants | This paper | https://github.com/JuliaFlynn /PacBio_barcode_assocation, (copy archived at swh:1:rev:29eac 92475a9ff8e24fb390986 c865b504c03f51) | See Materials and Methods section "Barcode Association" |
| Software, algorithm | GraphPad Prism 9 | Graphpad.com | RRID:SCR_008520 | |
| Software, algorithm | Flowjo v.10.8.0 | BD Biosciences | RRID:SCR_008520 | |
| Software, algorithm | Pymol v. 2.5.2 | Schrödinger | RRID:SCR_000305 | |
| Software, algorithm | MatPlotLib | http://matplotlib. sourceforge.net | RRID:SCR_008624 | |
| Sequence-based reagent | Sequencing primers | This paper | | See *Supplementary file 1* |
| Sequence-based reagent | Site-directed mutagenesis primers | This paper | | See *Supplementary file 1* |

## Construction of WT Ub-M$^{pro}$ vector (p416LexA_UbM$^{pro}$(WT)_B112)

The Ub-M$^{pro}$ gene fusion was constructed using overlapping PCR of the yeast Ub gene and SARS-CoV-2 M$^{pro}$ gene (*Jin et al., 2020*), and was inserted into the pRS416 vector after digestion with SpeI and BamHI. Four LexA boxes were amplified from the LexAbox4_citrine plasmid (FRP793_insul-(lexA-box)4-PminCYC1-Citrine-TCYC1 was a gift from Joerg Stelling; Addgene plasmid # 58434; http://n2t.net/addgene:58434; *Ottoz et al., 2014*) and inserted between the SacI and SpeI sites upstream of the Ub-M$^{pro}$ gene. The LexA_ER_B112 TF was amplified from Addgene_58437 (FRP880_PACT1(−1–520)-LexA-ER-haB112-TCYC1 was a gift from Joerg Stelling; Addgene plasmid # 58437; http://n2t.net/addgene:58437; *Ottoz et al., 2014*) and inserted into the KpnI site. The resulting vector is named (p416LexA-UbM$^{pro}$(WT)-B112). A destination vector was generated by removing the M$^{pro}$ sequence and replacing it with a restriction site for SphI.

## Generating mutant libraries

The SARS-CoV-2 M$^{pro}$ (ORF1ab polyprotein residues 3264–3569, GenBank code: MN908947.3) single site variant library was synthesized by Twist Biosciences (twistbioscience.com) by massively parallel oligonucleotide synthesis. In the library, each amino acid position was modified to all 19 amino acid variants plus a premature termination encoded by a stop codon, using the preferred yeast codon for each substitution. All 306 amino acids of M$^{pro}$ were modified yielding 6120 total variants. Due to challenges in construction, positions 27 and 28 were missing from the library. About 35 bp of sequence homologous to the destination vector was added to both termini of the library during synthesis to enable efficient cloning. The library was combined via Gibson assembly (NEB) with the destination vector. To avoid bottlenecking the library, sufficient transformations were performed to recover more than 50 independent transformants for each designed M$^{pro}$ variant in the library. To improve efficiency and accuracy of deep sequencing steps during bulk competition, each variant of the library was tagged with a unique barcode. A pool of DNA constructs containing a randomized 18 bp barcode sequence (N18) was cloned into the NotI and AscI sites

upstream of the LexA promoter sequence via restriction digestion, ligation and transformation into chemically competent *Escherichia coli*. These experiments were performed at a scale designed to have each M$^{pro}$ variant represented by 10–20 unique barcodes. The resulting library is named p416LexA-UbM$^{pro}$(lib)-B112.

## Barcode association

To associate barcodes with M$^{pro}$ variants, we digested the p416-UbM$^{pro}$(lib)-B112 plasmid upstream of the N18 sequence and downstream of the M$^{pro}$ sequence with NotI and SalI enzymes (NEB). The resulting 1800 bp fragment containing the barcoded library was isolated by Blue Pippen selecting for a 1–4 kB range. Of note, we determined it was important to avoid PCR to prepare the DNA for PacBio sequencing, as PCR led to up to 25% of DNA strands recombining, leading to widespread mismatch between the barcode and M$^{pro}$ variant. DNA was prepared for sequencing with the Sequel II Binding Kit v2.1 and the libraries were sequenced on a Pacific Biosciences Sequel II Instrument using a 15 hr data collection time, with a 0.4 hr pre-extension time (PacBio Core Enterprise, UMass Chan Medical School, Worcester, MA). PacBio circular consensus sequences were generated from the raw reads using SMRTLink v.10.1 and standard read-of-insert analysis parameters. After filtering low-quality reads (Phred scores<10), the data was organized by barcode sequence using custom analysis scripts that have been deposited on GitHub (https://github.com, see Key Resource Table). For each barcode that was read more than three times, we generated a consensus of the M$^{pro}$ sequence that we compared to WT to call mutations.

As a control for library experiments, the WT Ub-M$^{pro}$ gene was also barcoded with approximately 150 unique barcode sequences. The randomized 18 bp barcode sequence (N18) was cloned between the NotI and AscI sites upstream of the LexA promoter sequence in the p416LexA-Ub-M$^{pro}$(WT)-B112 vector with the goal of the WT sequence being represented by approximately 100 barcodes. The barcoded region of the plasmid was amplified by PCR using the primers listed in *Supplementary file 1* (for the WT barcoding it was not necessary to avoid strand recombination) and sequenced by EZ Amplicon deep sequencing (https://www.genewiz.com/).

## Generating split TF strain

The GFP reporter strain was generated by integration of GFP driven by a Gal1 promoter together with a HIS3 marker into the HO genomic locus. The Gal4, Gal80 and Pdr5 genes were disrupted to create the following strain: W303 *HO::Gal1-GFP-v5-His3; gal4::trp1; gal80::leu2 pdr5::natMX*.

The Gal4 DBD-M$^{pro}$CS-activation domain fusion gene (DBD-M$^{pro}$CS-AD) was generated by over-lapping PCR. The Gal4 DBD was amplified by PCR with a forward primer containing the EcoRI site and a reverse primer containing the extending M$^{pro}$CS overhang sequence. The Gal4 AD was amplified by PCR with a forward primer containing the M$^{pro}$CS overhang sequence and a reverse primer containing the SacI site (SacI_R). The DBD-M$^{pro}$CS-AD fusion gene was generated using the over-lapping DBD-M$^{pro}$CS and M$^{pro}$CS-AD products from above as templates and the EcoRI_F and SacI_R primers. The resulting DBD-M$^{pro}$CS-AD fusion gene was inserted between the EcoRI and SacI sites downstream of the CUP promoter in the integrative bidirectional pDK-ATC plasmid (kindly provided by D. Kaganovich; *Amen and Kaganovich, 2017*). The mCherry gene was subsequently cloned into the XhoI/BamHI sites downstream of the TEF promoter in the opposite orientation to create the plasmid pDK-CUP-DBD-M$^{pro}$CS-AD-TEF-mCherry. The fragment for genomic integration was generated by PCR with the primers listed in *Supplementary file 1*, was transformed into the reporter stain using LiAc/PEG transformation (*Gietz et al., 1995*), and successful integration of the module into the adenine biosynthesis gene was verified by PCR.

## Bulk split TF competition experiment

Barcoded WT UbM$^{pro}$ (p416LexA-UbM$^{pro}$(WT)-N18) plasmid was mixed with the barcoded UbM$^{pro}$ library (p416LexA-UbM$^{pro}$(lib)-N18) at a ratio of 20-fold WT to the average library variant. The blended plasmid library was transformed using the lithium acetate procedure into the reporter strain (*W303 ade::CUP-DBD-M$^{pro}$CS-AD-TEF-mCherry; ho::gal1-gfp-v5-his3; gal4::trp1; gal80::leu2; pdr5::natMX*). Sufficient transformation reactions were performed to attain about 5 million independent yeast transformants representing a 50-fold sampling of the average barcode. Each biological replicate represents a separate transformation of the library. Following 12 hr of recovery in synthetic dextrose lacking

adenine (SD-A), transformed cells were washed three times in synthetic dextrose lacking adenine and uracil (SD-A-U) media (SD-A-U to select for the presence of the M$^{pro}$ variant plasmid) to remove extra-cellular DNA and grown in 500 mL SD-A-U media at 30°C for 48 hr with repeated dilution to maintain the cells in log phase of growth and to expand the library. At least $10^7$ cells were passed for each dilution to avoid population bottlenecks. Subsequently, the library was diluted to early log phase in 100 mL of SD-A-U, grown for 2 hr, the culture was split in half, and 125 nM β-estradiol (from a 10 mM stock in 95% ethanol, Sigma-Aldirch) was added to one of the cultures to induce Ub-M$^{pro}$ expression. Cultures with and without β-estradiol were grown with shaking at 180 rpm for 6 hr at which point samples of ~$10^7$ cells were collected for FACS analysis.

## FACS sorting of TF screen yeast cells

A sample of $10^7$ cells were washed three times with 500 μL of tris-buffered saline containing 0.1% Tween and 0.1% bovine serum albumin (TBST-BSA). Cells were diluted to $10^6$ /mL and transferred to polystyrene FACS tubes. Samples were sorted for GFP and mCherry expression on a FACS Aria II cell sorter with all cells expressing cut TF (low GFP expression) in one population and uncut TF (high GFP expression) in a second population. To ensure adequate library coverage, we sorted at least 1.5 million cells of each population and collected them in SD-A-U media. For the first replicate, sorted yeast cells were amplified in 20 mL SD-U-A media for 10 hr at 30°C. These yeast samples were collected by centrifugation and cell pellets were stored at –80°C. It was observed that different populations of cells recovered at different rates during this amplification period, so in the second replicate cells were immediately spun down and stored at –80°C. Functional scores between the two replicates correlated well indicating that the amplification step was dispensable.

## Generating FRET strain

The YPet-CyPet FRET pair is a YFP-CFP fluorescent protein pair that has been fluorescently optimized by directed evolution for intracellular FRET (*Nguyen and Daugherty, 2005*). The YPet- M$^{pro}$CS-CyPet fusion gene was generated by overlapping PCR as follows. The CyPet gene was amplified by PCR from the pCyPet-His vector (pCyPet-His was a gift from Patrick Daugherty; Addgene plasmid #14030; http://n2t.net/addgene:14030) with a forward primer containing the BamHI site (BamHI_F) and a reverse primer containing the extending M$^{pro}$CS overhang sequence. The YPet gene was amplified by PCR from the pYPet-His vector (pYPet-His was a gift from Patrick Daugherty; Addgene plasmid #14031; http://n2t.net/addgene:14031) with a forward primer containing the extending M$^{pro}$CS over-hang sequence and a reverse primer containing the XhoI site (XhoI_R). The CyPet-M$^{pro}$CS-YPet fusion gene was generated using the overlapping CyPet-M$^{pro}$CS and M$^{pro}$CS-YPet products from above as templates and BamHI_F and XhoI_R primers. The resulting CyPet- M$^{pro}$CS-YPet gene was inserted between the BamHI and XhoI sites downstream of the TEF promoter in the integrative bidirectional pDK-ATG plasmid (kindly provided by D. Kaganovich; *Amen and Kaganovich, 2017*). The fragment for genomic integration was generated by PCR with the primers listed in *Supplementary file 1*, was transformed into W303 (*leu2-3,112 trp1-1 can1-100 ura3-1 ade2-1 his3-11,15*) using LiAc/PEG transformation (*Gietz et al., 1995*), and successful integration of the module into the adenine biosynthesis gene was verified by PCR.

## Bulk FRET competition experiment

The plasmid library including the barcoded WT plasmid was transformed as above using the lithium acetate procedure into *W303 Ade::TEF-CyPet-M$^{pro}$CS-YPet* cells. Sufficient transformation reactions were performed to attain about 5 million independent yeast transformants representing a 50-fold sampling of the average barcode. Cultures were grown and induced with β-estradiol as above for the TF screen with the exception that cells were induced for 1.5 hr. Samples of $10^7$ cells were collected for FACS analysis.

## FACS sorting of FRET screen yeast cells

A sample of $10^7$ cells were washed three times with 500 μL of TBST-BSA. Cells were diluted to $10^6$ / mL and transferred to polystyrene FACS tubes. Samples were sorted for YFP and CFP expression on a FACS Aria II cell sorter with all cells expressing cut FRET pair (low FRET) in one population and uncut FRET pair (high FRET) in a second population. To ensure adequate library coverage, we sorted at least

3 million cells of each population and collected them in SD-A-U media. Yeast samples were collected by centrifugation and cell pellets were stored at –80°C.

## Growth strain

The plasmid library including the barcoded WT plasmid was transformed as above using the lithium acetate procedure into W303 cells. Sufficient transformation reactions were performed to attain about 5 million independent yeast transformants representing a 50-fold sampling of the average barcode. Each biological replicate represents a separate transformation of the library. Following 12 hr of recovery in synthetic dextrose (SD) media, transformed cells were washed three times in SD-U media (SD lacking uracil to select for the presence of the $M^{pro}$ variant plasmid) to remove extracellular DNA and grown in 500 mL SD-U media at 30°C for 48 hr with repeated dilution to maintain the cells in log phase of growth ($OD_{600}$ = 0.05 – 1) and to expand the library. At least $10^7$ cells were passed for each dilution to avoid population bottlenecks. Subsequently, the library was diluted to early log phase ($OD_{600}$ = 0.05) in 100 mL of SD-U, grown for 2 hr, the culture was split in half, and 2 μM β-estradiol (from a 10 mM stock in 95% ethanol) was added to one of the cultures to induce Ub-$M^{pro}$ expression. Cultures with and without β-estradiol were grown with shaking at 180 rpm for 16 hr with dilution after 8 hr to maintain growth in exponential phase. Samples of ~$10^8$ cells were collected by centrifugation and cell pellets were stored at –80°C.

## DNA preparation and sequencing

We isolated plasmid DNA from each FACS cell population and the time points from the growth experiment as described (*Jiang et al., 2013*). Additionally, we sequenced the original barcoded plasmid library to evaluate the collateral effects on variants during the pre-selection library expansion stages. Purified plasmid DNA was linearized with AscI. Barcodes were amplified with 22 cycles of PCR using Phusion polymerase (NEB) and primers that add Illumina adapter sequences and a 6 bp identifier sequence used to distinguish cell populations. PCR products were purified two times over silica columns (Zymo Research) and quantified using the KAPA SYBR FAST qPCR Master Mix (Kapa Biosystems) on a Bio-Rad CFX machine. Samples were pooled and sequenced on an Illumina NextSeq instrument in single-end 75 bp mode.

## Analysis of Illumina sequencing data

We analyzed the Illumina barcode reads using custom scripts that have been deposited on GitHub (https://github.com, see Key Resource Table). Illumina sequence reads were filtered for Phred scores>10 and strict matching of the sequence to the expected template and identifier sequence. Reads that passed these filters were parsed based on the identifier sequence. For each screen/cell population, each unique N18 read was counted. The unique N18 count file was then used to identify the frequency of each mutant using the variant-barcode association table. To generate a cumulative count for each codon and amino acid variant in the library, the counts of each associated barcode were summed.

## Determination of functional scores

To determine the functional score for each variant in the two FACS-based screens, the fraction of each variant in the cut and uncut windows was first calculated by dividing the sequencing counts of each variant in a window by the total counts in that window. The functional score was then calculated as the fraction of the variant in the cut window divided by the sum of the fraction of the variant in the cut and uncut windows. The functional score for the growth screen was calculated by the fraction of the variant at the 0 hr time point divided by the sum of the fraction of the variant in the 0 and 16 hr time points. Functional scores were not calculated for variants with less than 100 total reads. The functional scores were normalized setting the score for the average WT $M^{pro}$ barcode as 1 and the average stop codon as 0. Both the unnormalized and normalized scores are reported in *Figure 2—source data 1*. For comparison, the counts for the growth-based screen were fit to selection coefficients (slope of $\log_2$(variant/WT counts)). We chose to report the functional scores as opposed to the selection coefficients in this paper so they would be directly comparable to the TF and FRET functional scores.

## Analysis of $M^{pro}$ expression and Ub removal by Western blot

To facilitate analysis of expression levels of $M^{pro}$ and examine effective removal of Ub, a his tag was fused to the C-terminus of $M^{pro}$ to create the plasmid p416LexA-Ub$M^{pro}$-his$_6$-B112. In addition, the

C145A mutation was created by site-directed mutagenesis to ensure cleavage by Ub specific proteases and to reduce the toxicity caused by WT M$^{pro}$ expression. W303 cells were transformed with the p416LexA-UbM$^{pro}$(C145A)-his$_6$ construct and the resulting yeast cells were grown to exponential phase in SD-U media at 30°C. 2 μM β-estradiol was added when indicated and cells were grown for an additional 8 hr. About $10^8$ yeast cells were collected by centrifugation and frozen as pellets at −80°C. Cells were lysed by vortexing the thawed pellets with glass beads in lysis buffer (50 mM Tris-HCl pH 7.5, 5 mM EDTA and 10 mM PMSF), followed by addition of 2% sodium dodecyl sulfate (SDS). Lysed cells were centrifuged at 18,000 g for 1 min to remove debris, and the protein concentration of the supernatants was determined using a BCA protein assay kit (Pierce) compared to a bovine serum albumin (BSA) protein standard. Around 15 μg of total cellular protein was resolved by SDS-PAGE, transferred to a PVDF membrane, and probed using an anti-his HRP-conjugated antibody (R&D systems). Purified M$^{pro}$-his$_6$ protein was a gift from the Schiffer laboratory. There is a slight size difference on the Western blot between the purified M$^{pro}$-his$_6$ protein and the C145A M$^{pro}$-his$_6$ in the yeast lysate. We do not completely understand the origin of this mobility shift, but possible causes are an abnormal gel shift due to the C145A mutation, a mobility difference due to buffer, nucleic acids or additional proteins in the lysate, or an unknown modification of M$^{pro}$ in bacteria compared to yeast.

## Sequence and structure analysis

Evolutionary conservation was calculated with an alignment of homologs from diverse species using the ConSurf server (*Ashkenazy et al., 2016*). The effects of single mutations on protein-ligand interactions were predicted by calculating the binding affinity changes using PremPLI (https://lilab.jysw.suda.edu.cn/research/PremPLI/; *Sun et al., 2021*). The figures were generated using Matplotlib (*Hunter, 2007*), PyMOL and GraphPad Prism version 9.3.1.

## Identifying mutations in circulating SARS-COV-2 sequences

The complete set of SARS-COV-2 isolate genome sequences was downloaded from the GISAID database. The SARS-COV-2 M$^{pro}$ reference sequence (NCBI accession NC_045512.2) was used as a query in a tBLASTn search against the translated nucleotide sequences of these isolates to identify the M$^{pro}$ region and its protein sequence for each isolate, if present. M$^{pro}$ sequences were discarded if they contained 10 or more ambiguous 'X' amino acids or had amino acid length less than 290. A multiple sequence alignment was performed and for each of the twenty standard amino acids, the number of times it was observed at each position in the M$^{pro}$ sequence was calculated.

## Acknowledgements

This work was sponsored by Novartis Institutes for BioMedical Research. We would like to thank the UMass Chan Medical School Pacific Biosciences Core Enterprise for providing the PacBio NGS services and the UMass Chan Medical School Flow Cytometry Core Facility for providing the FACS services. We would also like to thank Ala Shaqra, Sarah Zvornicanin, and Qiu Yu Huang for conceptual discussions of the manuscript.

## Additional information

### Competing interests

David T Barkan, Stephanie A Moquin, Dustin Dovala: is an employee of Novartis. The other authors declare that no competing interests exist.

## Funding

| Funder | Grant reference number | Author |
|---|---|---|
| Novartis Institutes for BioMedical Research | | Julia M Flynn<br>Neha Samant<br>Gily Schneider-Nachum<br>Nese Kurt Yilmaz<br>Celia A Schiffer<br>Daniel NA Bolon |

The funders had no role in study design, data collection and interpretation, or the decision to submit the work for publication.

## Author contributions

Julia M Flynn, Conceptualization, Investigation, Methodology, Project administration, Validation, Visualization, Writing - original draft, Writing – review and editing; Neha Samant, Gily Schneider-Nachum, Conceptualization, Investigation, Methodology, Writing – review and editing; David T Barkan, Conceptualization, Data curation, Formal analysis; Nese Kurt Yilmaz, Conceptualization, Investigation, Writing – review and editing; Celia A Schiffer, Conceptualization, Funding acquisition, Project administration, Writing – review and editing; Stephanie A Moquin, Conceptualization, Methodology, Resources, Writing – review and editing; Dustin Dovala, Conceptualization, Methodology, Project administration, Resources, Validation, Writing – review and editing; Daniel NA Bolon, Conceptualization, Funding acquisition, Methodology, Project administration, Writing - original draft, Writing – review and editing

## Author ORCIDs

Julia M Flynn (iD) http://orcid.org/0000-0002-5490-393X
Neha Samant (iD) http://orcid.org/0000-0002-9514-0705
Gily Schneider-Nachum (iD) http://orcid.org/0000-0003-3028-350X
Celia A Schiffer (iD) http://orcid.org/0000-0003-2270-6613
Daniel NA Bolon (iD) http://orcid.org/0000-0001-5857-6676

## Decision letter and Author response

Decision letter https://doi.org/10.7554/eLife.77433.sa1
Author response https://doi.org/10.7554/eLife.77433.sa2

---

# Additional files

## Supplementary files

• Supplementary file 1. List of oligomers used in this study.
• Transparent reporting form

## Data availability

Next generation sequencing data has been deposited to the NCBI short read archive (PRJNA842255). Tabulated raw counts of all variants in all replicates are included in Figure 2 - source data 1. Figure 2 - source data 1, Figure 4 - source data 1, Figure 4 - source data 2, and Figure 5 - source data 1 contain the data used to generate all the figures.

The following dataset was generated:

| Author(s) | Year | Dataset title | Dataset URL | Database and Identifier |
|---|---|---|---|---|
| Flynn JM, Bolon DNA | 2022 | Comprehensive fitness landscape of SARS-CoV-2 Mpro in *S. cerevisiae* - raw sequence reads | https://www.ncbi.nlm.nih.gov/bioproject/?term=PRJNA842255 | NCBI BioProject, PRJNA842255 |

---

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
