## [Editor Report]

This manuscript utilizes modern molecular tools to construct a fitness landscape in SARS-CoV-2, yielding insight into potential resistance mechanisms. The paper is rigorous, well-written, and has very clear implications in the biomedical realm.

---

## [Decision Letter]

**Decision letter after peer review:**

Thank you for submitting your article "Comprehensive fitness landscape of SARS-CoV-2M^pro^ reveals insights into viral resistance mechanisms" for consideration by *eLife*. Your article has been reviewed by 2 peer reviewers, and the evaluation has been overseen by a Reviewing Editor and Naama Barkai as the Senior Editor. The following individuals involved in the review of your submission have agreed to reveal their identity: James S Fraser (Reviewer #1); Daniel Weinreich (Reviewer #2).

Essential revisions:

1) Both authors had editorial and textual changes that require attention. While not major, they need to be addressed in a revision and highlighted in a revised text.

2) In addition, there were clarifying questions about some of the visualizations and explanations that need to be addressed in a detailed revision letter and fixed in a revised manuscript. While these are not major technical points they should nonetheless be addressed.

*Reviewer #2 (Recommendations for the authors):*

Paragraph beginning line 41 describes some amazing biology. A few more citations might be helpful.

Line 69-70: are there 2 catalytic sites in the dimer?

Line 86-87: more than just biophysical properties, right?

Line 105: maybe one clause on yeast toxicity here? It becomes clear shortly, but the question will arise in some (other) readers' minds at this point.

Lines 131-141: there's a lot going on to make these assays work. Very impressive/very well done!

(Second!) Figure 1 (p. 8), panel A: left lane labeled "pure protein" is what's called the control in the legend, right? Standardize language? Also, sizes in 1st and 3rd lanes don't quite align; why?

Line 183: no synonymous mutations, which other authors have used as noise control. Why doesn't this matter?

Figure 2A, 3rd panel: what explains the curvature in the data? 2C: lovely that the stop codons at the end don't affect function.

Figure 3: sexy but lots of real estate. Anyone who needs that level of detail will go to the source data. (Which, incidentally is labeled "Figure 2 – source data" in the legend.) Perhaps suppress?

Figure 4: why are the inter-assay regressions linear, yet the regressions to the catalytic assays bend by different amounts in the 3 panels in the bottom row?

Line 256: why not also follow through with TF functional scores?

Line 262: what's the difference between hCoV-19 and SARS-CoV-2? Why the shift in nomenclature?

Line 270: the 290 non-synonymous mutations observed more than 100 times IN ISOLATION or together with other mutations? How does the authors' analysis deal with epistasis among mutations seen in clinical isolates?

Line 311: what is an "allosteric" communication network? I thought allostery meant structural response on binding another ligand. Yes, there may be some mechanical/physical linkage between sites outside the pocket, but here the linkage seems distinct from allostery.

Line 328-340: How does this affect drug design strategies? Is conservation enough or must we target functional hotspots?

Line 344-5; why focus on mutations that are compatible with function, and available to drug resistance? Why not on those that are incompatible with function? Somehow this must connect with substrate envelope-thinking, but how? Linkage is obscure to me.

---

## [Author Response]

Reviewer #2 (Recommendations for the authors):Paragraph beginning line 41 describes some amazing biology. A few more citations might be helpful.

We have added citations as suggested.

Line 69-70: are there 2 catalytic sites in the dimer?

Yes. We modified the text to state “catalytic dyads”

Line 86-87: more than just biophysical properties, right?

Yes, we updated the text to add biochemical.

Line 105: maybe one clause on yeast toxicity here? It becomes clear shortly, but the question will arise in some (other) readers' minds at this point.

We have updated the text as suggested.

Lines 131-141: there's a lot going on to make these assays work. Very impressive/very well done!(Second!) Figure 1 (p. 8), panel A: left lane labeled "pure protein" is what's called the control in the legend, right? Standardize language?

Yes, we changed the label to control on gel.

Also, sizes in 1st and 3rd lanes don't quite align; why?

We added the following to Materials and methods:

“There is a slight size difference on the Western blot between the purified M^pro^-his_6_ protein and the C145A M^pro^-his_6_ in the yeast lysate. We do not completely understand the origin of this mobility shift, but possible causes are an abnormal gel shift due to the C145A mutation, a mobility difference due to buffer, nucleic acids or additional proteins in the lysate, or an unknown modification of M^pro^ in bacteria compared to yeast.”

Line 183: no synonymous mutations, which other authors have used as noise control. Why doesn't this matter?

We include multiple barcodes of wild-type in these experiments instead of synonyms. We have used the distributions of stops and wild-type barcodes as indications of noise.

Figure 2A, 3rd panel: what explains the curvature in the data?

It is likely due to a slight difference in induction of M^pro^ expression between the two replicates. For instance, a slight difference in concentration or strength of β-estradiol used on different days would cause a slight difference in M^pro^ expression and thus in toxicity.

2C: lovely that the stop codons at the end don't affect function.Figure 3: sexy but lots of real estate. Anyone who needs that level of detail will go to the source data. (Which, incidentally is labeled "Figure 2 – source data" in the legend.) Perhaps suppress?

In our experience, there remain a large fraction of readers that appreciate having the heatmaps. We have moved two of the heatmaps to the supplement and retained the FRET map in Figure 3.

Figure 4: why are the inter-assay regressions linear, yet the regressions to the catalytic assays bend by different amounts in the 3 panels in the bottom row?

While it was difficult to tell in the original figure, the inter-assay regressions also bend due to the fact that the average defective mutation is more exaggerated in the growth screen compared to the TF and FRET screen. We have made this easier to judge by adding a dashed line corresponding to the diagonal in Figure 4C.

Line 256: why not also follow through with TF functional scores?

We have added all the TF data analyses to the supplement figures.

Line 262: what's the difference between hCoV-19 and SARS-CoV-2? Why the shift in nomenclature?

They are the same thing, the hCoV-19 nomenclature was used to denote the clinical variants in human samples, but we have changed this to human SARS-CoV-2 to be clearer and more consistent.

Line 270: the 290 non-synonymous mutations observed more than 100 times IN ISOLATION or together with other mutations? How does the authors' analysis deal with epistasis among mutations seen in clinical isolates?

The vast majority of mutations observed in this set of variants were in isolation. We have added the following to the text:

“The vast majority of the clinical isolates that have been sequenced to date have either 0 or 1 M^pro^ mutations with fewer than 0.4% having 2 or greater mutations and thus we did not account for epistasis in our analysis.”

Line 311: what is an "allosteric" communication network? I thought allostery meant structural response on binding another ligand. Yes, there may be some mechanical/physical linkage between sites outside the pocket, but here the linkage seems distinct from allostery.

We changed the terminology to distal regulatory communication network.

Line 328-340: How does this affect drug design strategies? Is conservation enough or must we target functional hotspots?

We added to the main text “It is thus important when designing drugs to disrupt M^pro^ function to focus on the functionally critical sites which are a subset of the evolutionarily conserved positions.”

Line 344-5; why focus on mutations that are compatible with function, and available to drug resistance? Why not on those that are incompatible with function? Somehow this must connect with substrate envelope-thinking, but how? Linkage is obscure to me.

Mutations that are not compatible with function will be unlikely to evolve because M^pro^ function is required for viral propagation.